# Personalized Federated Conformal Prediction with Localization

**Yinjie Min**
School of Statistics and Data Science
Nankai University
nk.yjmin@gmail.com

**Chuchen Zhang**
School of Statistics and Data Science
Nankai University
chuchenz@mail.nankai.edu.cn

**Liuhua Peng** [†]
School of Mathematics & Statistics
The University of Melbourne
liuhua.peng@unimelb.edu.au

**Changliang Zou** [†]
School of Statistics and Data Science, LPMC and KLMDASR and LEBPS
Nankai University
nk.chlzou@gmail.com

## Abstract

Personalized federated learning addresses data heterogeneity across distributed agents but lacks uncertainty quantification that is both agent-specific and instance-specific, which is a critical requirement for risk-sensitive applications. We propose personalized federated conformal prediction (PFCP), a novel framework that combines personalized federated learning with conformal prediction to provide statistically valid agent-personalized prediction sets with instance-localization. By leveraging privacy-preserving knowledge transfer from other source agents, PFCP ensures marginal coverage guarantees for target agents while significantly improving conditional coverage performance on individual test instances, which has been validated by extensive experiments.

## 1 Introduction

The rise of distributed intelligent systems presents a fundamental challenge: heterogeneous data distributions across agents necessitate personalized modeling and analysis to capture agent-specific characteristics. In domains such as medical diagnostics [41] and autonomous driving [9], agent-specific feature-label relationships can differ substantially. These discrepancies often render global models ineffective, motivating the development of personalized federated learning (PFL) [37]. Unlike conventional federated learning, which prioritizes a unified global model, PFL explicitly balances shared knowledge transfer with **agent-specific** adaptation, enabling personalized models that respect data heterogeneity while preserving privacy.

In risk-sensitive domains, agent-specific modeling is insufficient. Safety-critical decisions, such as those made by autonomous vehicles under sensor noise or clinical models diagnosing rare patient subgroups, require reliable uncertainty quantification to ensure safety [19]. Conformal prediction (CP) offers a statistically valid approach, providing set-valued outputs with rigorous marginal coverage

---

[†]Corresponding author

39th Conference on Neural Information Processing Systems (NeurIPS 2025).

guarantees. In this paper, we incorporate conformal prediction into the PFL framework for uncertainty quantification to ensure prediction sets remain valid at the individual agent level.

Given a calibration dataset $\{(X_i, Y_i)\}_{i=1}^n$ from a joint distribution $P$, conformal prediction constructs a set $\widehat{C}_\alpha(X_{n+1})$ for a test instance $X_{n+1}$ at a desired confidence level $1 - \alpha \in (0, 1)$ such that

$$\mathbb{P}\big(Y_{n+1} \in \widehat{C}_\alpha(X_{n+1})\big) \geq 1 - \alpha, \tag{1}$$

providing distribution-free coverage guarantees without assumptions on $P$ [34]. Split conformal prediction (SCP) [39], a widely used variant of conformal prediction, achieves marginal coverage by leveraging pretrained models and calibration data to compute conformity scores.

Marginal coverage (1) ensures average validity over $X_{n+1}$ but can mask severe miscoverage for individual instances. In medical diagnosis, undercoverage for patients with high-risk disease features, such as rare genetic markers or specific symptom combinations [19], can lead to delayed diagnoses and life-threatening consequences. A more meaningful guarantee is the **test-conditional coverage**:

$$\mathbb{P}(Y_{n+1} \in \widehat{C}_\alpha(X_{n+1}) \mid X_{n+1} = x) \geq 1 - \alpha$$

for all $x$, which aligns prediction sets with local uncertainty. This motivates the design of conformal prediction methods with **instance-specific** adaptation, which conventional CP and SCP do not address.

However, achieving such conditional coverage is generally impossible without strong distributional assumptions [21]. Several conformal prediction methods have been developed that aim for approximate or asymptotic test-conditional coverage by introducing localization and instance-specific adaptation. These approaches either modify the calibration step [25, 26, 20, 13, 12] or employ alternative score functions [32, 5, 14, 6]. Several of these approaches rely on estimating local distributional features, such as conditional quantiles [12] or conditional distributions [5, 13], which requires more calibration data to efficiently estimate local properties for instance-specific adaptation.

**Problem formulation.** Suppose there are $K + 1$ agents, each with a dataset $\mathcal{D}_k = \{Z_{k,i} = (X_{k,i}, Y_{k,i})\}_{i=1}^{n_k}$ containing $n_k$ independent samples from distribution $P_k$ supported on $\mathcal{X} \times \mathcal{Y}$. Without loss of generality, we designate agent $K + 1$ as the target agent. For simplicity, we omit the subscript $K + 1$ and denote $P_{K+1} =: P$ and $\mathcal{D}_{K+1} =: \mathcal{D} = \mathcal{D}_{\text{tr}} \cup \mathcal{D}_{\text{cal}}$, where $\mathcal{D}_{\text{tr}} = \{Z_i' = (X_i', Y_i')\}_{i=1}^n$ and $\mathcal{D}_{\text{cal}} = \{Z_i = (X_i, Y_i)\}_{i=1}^n$ are training and calibration sets of equal size $n$, drawn from $P = P_X \times P_{Y|X}$. Our goal is to construct a prediction set $\widehat{C}_\alpha(X_{n+1})$ for a test input $X_{n+1}$ from the target agent under the collaboration of the other $K$ agents without sharing raw data. Due to heterogeneity across agents ($P_k$ may differ) and variation in the conditional distribution of $Y$ given $X$ within agents, the prediction set should account for both inter-agent and intra-agent heterogeneity.

A natural solution is to apply distributional [5] or localized conformal prediction [13] using only the target agent's data. While this ensures marginal coverage and allows for instance-specific localization, it overlooks valuable information from other agents and may lead to unreliable prediction sets with poor local or instance-specific performance, especially in regions with limited or sparse calibration data, where estimating local distributional properties is challenging.

Federated learning (FL) enables collaborative model training across agents without sharing raw data, but conventional FL often produces a global model, neglecting agent-specific adaptation. Conformal prediction adds further challenges under FL due to privacy constraints and distributional heterogeneity, which violate conformal prediction's exchangeability assumption. Existing federated conformal prediction methods usually assume the test input $X_{n+1}$ is from a mixture distribution $P_w = \sum_{k=1}^K w_k P_k$ with $w_k \propto (n_k + 1)$, misaligned with agent-specific prediction and lacking marginal guarantees for the target agent. To bridge this gap, we consider constructing conformal prediction sets for a target agent using all agents' data in a privacy-preserving way, ensuring marginal coverage while achieving conditional coverage closer to the nominal level $1 - \alpha$ and producing narrower prediction sets compared to those constructed without using other agents.

**Contributions.** In this paper, we propose Personalized Federated Conformal Prediction (PFCP), a novel framework that constructs prediction sets with both agent-specific and instance-specific adaptation. The contributions of this paper are:

- We formalize generalized localized conformal prediction (GLCP), a novel extension of localized conformal prediction [13] that adopts any conditional distribution estimator and thus accommodates

any data type. GLCP enjoys marginal coverage guarantee and localization due to the transformed score that contains conditional property.

- We propose personalized federated conformal prediction (PFCP) framework with localization, which achieves marginal coverage for target agent under heterogeneity across agents. More importantly, PFCP incorporates data from other source agents to improve the local performance of the prediction sets under the federated learning framework without sharing raw data.

- We establish finite sample marginal coverage guarantee of PFCP, and theoretical non-asymptotic test-conditional coverage improvement of PFCP through cross-agent knowledge transfer compared with GLCP based solely on target agent data.

- We validate PFCP performance through synthetic and real-world dataset experiments, showing PFCP simultaneously maintains marginal coverage, improves test-conditional coverage performance significantly and reduces prediction set size compared to GLCP, which relies solely on scarce calibration and training data from the target agent.

**Related work.** Lu and Kalpathy-Cramer [22] first explore conformal prediction in the FL, proposing an aggregation algorithm for prediction set construction, albeit without formal coverage guarantees. Humbert et al. [16, 17] introduce FedCP-QQ, a one-shot federated conformal prediction. It computes the $1 - \alpha'$ quantile locally and the $1 - \beta'$ quantile of the $K$ local quantiles on a central server, providing $1 - \alpha$ marginal coverage with specified $\alpha'$ and $\beta'$ under the i.i.d. assumption across agents.

Another line of work, including Lu et al. [23], assumes test data are from the mixture distribution $P_w = \sum_{k=1}^{K} w_k P_k$. In their FedCP method, the central server aggregates sketch of scores from local agents to determine the quantile of conformal scores, ensuring marginal coverage under the FL exchangeability assumption. Wen et al. [40] extend this to a decentralized setting with limited calibration data and local communication, while Kang et al. [18] propose a robust version of FedCP under Byzantine attacks. However, these methods assume test data originate from a mixture distribution, making them unsuitable for personalized settings. In our PFL setting where test data come from a specific agent, the FL exchangeability [23] assumption is violated, and marginal coverage guarantees may no longer hold for FedCP and its variants.

The most closely related works to ours are Plassier et al. [27, 28], whose methods, referred to as CPlab and CPhet respectively, consider agent heterogeneity arising from covariate or label shift. They estimate global and target-agent densities to compute a density ratio, then apply weighted conformal prediction [38] to construct the prediction set. However, their marginal coverage depends heavily on accurate density ratio estimation, limiting robustness. In contrast, our PFCP method guarantees marginal coverage irrespective of estimator quality. Notably, all existing federated conformal prediction approaches have not considered localization or improvements in conditional coverage and size of prediction sets.

## 2 Methodology

Let $s(x, y)$ and $\{s_k(x, y)\}_{k=1}^{K}$ be pretrained conformity score functions for target agent and source agents. They measure how well a label $y$ aligns with the predicted value at $x$, with smaller values indicating higher conformity. Let $S_{k,i} = s_k(X_{k,i}, Y_{k,i})$ for $i \in [n_k]$ be scores from source agent $k$, and let $S_i = s(X_i, Y_i)$ and $S_i' = s(X_i', Y_i')$ for $i \in [n]$ be scores from target agent. For a test point $X_{n+1} \sim P_X$ from the target agent and a trial label $y$, define $S_i^y = S_i$ for $i \in [n]$, and $S_{n+1}^y = s(X_{n+1}, y)$.

Let $f(x, s)$ and $f_k(x, s)$ denote the joint density functions of $(X_1, S_1)$ and $(X_{k,1}, S_{k,1})$, respectively. Denote the marginal covariate density under the target distribution $P$ by $f(x)$ and under $P_k$ by $f_k(x)$ for $k \in [K]$. Let $f(s \mid x) = f(x, s)/f(x)$ denote the conditional PDF of $S_1$ given $X_1 = x$, and $F(s \mid x)$ the corresponding conditional CDF.

### 2.1 Generalized localized conformal prediction based solely on target agent data

In this section, we propose the Generalized Localized Conformal Prediction (GLCP) framework, which relies solely on target agent data. To account for heterogeneity in $P_{Y|X}$ and capture local information at a given point $x$, we estimate the conditional distribution of the score $S_1$ given $X_1 = x$

using dataset $\widetilde{\mathcal{D}}_{\mathrm{tr}} = \{(X'_i, S'_i)\}_{i=1}^n$. Let $\widehat{F}(s \mid x) = \mathcal{A}(\widetilde{\mathcal{D}}_{\mathrm{tr}})$ be the estimator of $F(s \mid x)$, where $\mathcal{A}(\cdot)$ represents a generic conditional distribution estimation algorithm, e.g., engression in Shen and Meinshausen [35]. Define $V_i^y = \widehat{F}(S_i^y \mid X_i)$ for $i \in [n+1]$. The GLCP set is defined as

$$\widehat{C}_\alpha^{\mathrm{GLCP}}(X_{n+1}) = \Big\{ y : V_{n+1}^y \leq Q\Big(1 - \alpha; (n+1)^{-1}\Big(\sum_{i=1}^n \delta_{V_i^y} + \delta_1\Big)\Big) \Big\}, \qquad (2)$$

where $\delta_v$ denotes the point mass at $v$ and $Q(1 - \alpha; \cdot)$ denotes the $(1 - \alpha)$-quantile of the distribution in the second argument. The following lemma follows directly from the basic validity guarantees of conformal prediction theory [39].

**Lemma 2.1.** *If data pairs* $(X_1, Y_1), \ldots, (X_n, Y_n), (X_{n+1}, Y_{n+1})$ *are i.i.d. from P, and* $\widehat{F}(s \mid x)$ *is trained using dataset* $\widetilde{\mathcal{D}}_{\mathrm{tr}} = \{(X'_i, S'_i)\}_{i=1}^n$, *we have* $\mathbb{P}\big(Y_{n+1} \in \widehat{C}_\alpha^{\mathrm{GLCP}}(X_{n+1})\big) \geq 1 - \alpha$.

Actually, GLCP is designed to *generalize* and *unify* existing approaches. If we choose $\widehat{F}(s \mid x)$ to be the Nadaraya–Watson estimator [15], GLCP reduces to the localized conformal prediction in Guan [13]. If the score function is specified as $s(x, y) = y$, GLCP becomes equivalent to distributional conformal prediction in Chernozhukov et al. [5]. Notably, GLCP accommodates any conditional distribution estimator and thus accommodates any data type, providing greater flexibility. For instance, one may adopt estimators tailored to high-dimensional covariates, thereby alleviating the curse of dimensionality inherent to the Nadaraya–Watson estimator used in Guan [13].

GLCP naturally supports localization and instance-specific adaptation, emerging seamlessly from its formulation. Given a score function $s(\cdot, \cdot)$, the oracle conditional conformal prediction set is defined as $\{y : s(x, y) \leq Q(1 - \alpha, F(s \mid x))\}$, which is equivalent to $\{y : F(s(x, y) \mid x) \leq 1 - \alpha\}$. The transformed score $V_i^y$ serves as an estimator of $F(S_i^y \mid X_i)$, and it converges in distribution to $U[0, 1]$ if $\widehat{F}(s \mid x)$ is a consistent estimator of $F(s \mid x)$. In this case, the empirical quantile of $\{V_i^y\}_{i=1}^{n+1}$ converges to $1 - \alpha$, which ensures that the GLCP set $\widehat{C}_\alpha^{\mathrm{GLCP}}(X_{n+1})$ in (2) approximates the oracle conditional prediction set. Therefore, the GLCP inherits the asymptotic conditional property.

However, accurately estimating $F(s \mid x)$ at each $x \in \mathcal{X}$ requires a large sample size, and the estimation complexity grows with the dimensionality of the covariates. When the target agent has limited data, GLCP may produce unreliable or excessively large prediction sets in certain regions.

## 2.2 Personalized federated conformal prediction

In federated learning, directly pooling data across agents to construct prediction sets is infeasible due to privacy constraints and data heterogeneity, which violates exchangeability required by conformal prediction. Existing federated conformal prediction methods either assume no heterogeneity among agents or the test data are from a mixture distribution over all agents, which is incompatible with our personalized setting. Moreover, these approaches do not account for localization or instance-specific adaptation. To bridge this gap, we propose Personalized Federated Conformal Prediction (PFCP).

Define weight $\pi_k = n_k / \sum_{\ell=1}^K n_\ell$ for $k \in [K]$ and the mixture density as $f_{\mathrm{mix}}(x, s) = \sum_{k=1}^K \pi_k f_k(x, s)$ with corresponding marginal covariate density $f_{\mathrm{mix}}(x)$. Let $f_{\mathrm{mix}}(s \mid x) = f_{\mathrm{mix}}(x, s)/f_{\mathrm{mix}}(x)$, with corresponding CDF $F_{\mathrm{mix}}(s \mid x)$. Define the density ratios as $r(x) = f(x)/f_{\mathrm{mix}}(x)$, $r(x, s) = f(x, s)/f_{\mathrm{mix}}(x, s)$ and $r(s \mid x) = f(s \mid x)/f_{\mathrm{mix}}(s \mid x)$. The key idea of PFCP is to derive an enhanced estimator of the conditional distribution $F(s \mid x)$ that will be utilized in the transformed score $V_i^y$ by strategically incorporating auxiliary data from other agents. Regarding the following equivalent reformulations of $F(s \mid x)$:

$$F(s \mid x) = \mathbb{E}_{S \sim F(s \mid x)} \mathbb{1}(S \leq s) = \int_{-\infty}^s f(t \mid x) dt = \int_{-\infty}^s f_{\mathrm{mix}}(t \mid x) r(t \mid x) dt$$

$$= \mathbb{E}_{S \sim F_{\mathrm{mix}}(s \mid x)} \big\{ \mathbb{1}(S \leq s) r(S \mid x) \big\}. \qquad (3)$$

Therefore, we can derive an alternative estimator for $F(s \mid x)$ using estimators of $F_{\mathrm{mix}}(s \mid x)$ and corresponding density ratio $r(s \mid x)$ via (3). While $F(s \mid x)$ is estimated locally using algorithm $\mathcal{A}(\cdot)$ in Section 2.1, estimating $F_{\mathrm{mix}}(s \mid x)$ requires datasets from multiple source agents under a federated setting. This is achieved using algorithm $\widetilde{\mathcal{A}}(\cdot)$, which integrates local applications of algorithm $\mathcal{A}(\cdot)$ on each source agent locally with a federated learning algorithm (e.g., federated averaging [24]) as

detailed in Section A.1.1 of the Appendix. On the other hand, estimating $r(s \mid x)$ typically requires computing joint and marginal density ratios first, and taking their quotient afterwards, which may introduce errors. To circumvent this, we propose using a weight function $w(x)$ to aggregate the two $F(s \mid x)$ estimators in (5), bypassing $r(s \mid x)$ estimation while using an estimator of $r(x, s)$ to formulate the prediction set. The estimation of $r(x, s)$ is derived using algorithm $\widetilde{\mathcal{A}}_{\mathrm{dr}}(\cdot)$, which is the federated implementation of a density ratio estimation algorithm as detailed in Section A.1.2. Finally, we introduce $\mathcal{A}_{\mathrm{agg}}(\cdot)$ as the aggregation algorithm based on (5) that combines two estimators of the conditional distribution $F(s \mid x)$ and the estimator of the density ratio $r(x, s)$.

Due to space constraints, we present details of $\mathcal{A}(\cdot)$ (uses local data at target agent) and $\widetilde{\mathcal{A}}(\cdot)$ (uses data from $K$ source agents) in Section A.1.1 of the Appendix. The privacy-preserving density ratio estimation algorithm $\widetilde{\mathcal{A}}_{\mathrm{dr}}(\cdot)$ uses data from both target and source agents is detailed in Section A.1.2 of the Appendix. The aggregation algorithm $\mathcal{A}_{\mathrm{agg}}(\cdot)$ used to derive the enhanced estimator of the conditional distribution $F(s \mid x)$ is detailed in Section 2.3. Let $\widetilde{\mathcal{D}}_k = \{(X_{k,i}, S_{k,i})\}_{i=1}^{n_k}$ for $k \in [K]$. The PFCP procedure is summarized as below:

Step 1. Train $\widehat{F}(s \mid x) = \mathcal{A}(\widetilde{\mathcal{D}}_{\mathrm{tr}})$ as an estimator of $F(s \mid x)$ using $\widetilde{\mathcal{D}}_{\mathrm{tr}}$ in the target agent. In addition, train $\widehat{F}_{\mathrm{mix}}(s \mid x) = \widetilde{\mathcal{A}}(\widetilde{\mathcal{D}}_1, \ldots, \widetilde{\mathcal{D}}_K)$ as an estimator of $F_{\mathrm{mix}}(s \mid x)$ using $\widetilde{\mathcal{D}}_1, \ldots, \widetilde{\mathcal{D}}_K$ from $K$ source agents under the FL setting.

Step 2. Train $\widehat{r}(x, s) = \widetilde{\mathcal{A}}_{\mathrm{dr}}(\widetilde{\mathcal{D}}_{\mathrm{tr}}, \widetilde{\mathcal{D}}_1, \ldots, \widetilde{\mathcal{D}}_K)$ as an estimator of $r(x, s)$ under the FL setup using $\widetilde{\mathcal{D}}_{\mathrm{tr}}, \widetilde{\mathcal{D}}_1, \ldots, \widetilde{\mathcal{D}}_K$ from both target and source agents.

Step 3. Use an aggregation algorithm that combines $\widehat{F}(s \mid x)$, $\widehat{F}_{\mathrm{mix}}(s \mid x)$ and $\widehat{r}(x, s)$ to produce $\widehat{F}_{\mathrm{agg}}(s \mid x) = \mathcal{A}_{\mathrm{agg}}(\widehat{F}(s \mid x), \widehat{F}_{\mathrm{mix}}(s \mid x), \widehat{r}(x, s))$ as an enhanced estimator of $F(s \mid x)$.

Step 4. Define transformed scores on $\mathcal{D}_{\mathrm{cal}} \cup \{(X_{n+1}, y)\}$ as $V_{\mathrm{agg},i}^y = \widehat{F}_{\mathrm{agg}}(S_i^y \mid X_i)$ for $i \in [n+1]$. The PFCP set is then defined as

$$\widehat{C}_\alpha^{\mathrm{PFCP}}(X_{n+1}) = \left\{ y : V_{\mathrm{agg},n+1}^y \le Q\Big(1 - \alpha; (n+1)^{-1}\Big(\sum_{i=1}^n \delta_{V_{\mathrm{agg},i}^y} + \delta_1\Big)\Big)\right\}. \quad (4)$$

The algorithm following these steps is provided in Algorithm 5 in Section A.1.4 of the Appendix.

In (4), only data from the target agent is used for calibration, which guarantees marginal coverage regardless of the data heterogeneity across agents. Furthermore, while $\widehat{F}(s \mid x)$ trained on limited target data often performs poorly, the aggregated estimator $\widehat{F}_{\mathrm{agg}}(s \mid x)$ typically achieves higher accuracy. By substituting $\widehat{F}(s \mid x)$ with $\widehat{F}_{\mathrm{agg}}(s \mid x)$, we obtain superior estimates that simultaneously improve conditional coverage accuracy and yield tighter prediction sets.

Notably, $\widetilde{\mathcal{A}}(\cdot)$ and $\widetilde{\mathcal{A}}_{\mathrm{dr}}(\cdot)$ are federated learning algorithms involving data communication among agents, while $\mathcal{A}(\cdot)$ and $\mathcal{A}_{\mathrm{agg}}(\cdot)$ execute locally. Specifically, $\mathcal{A}(\cdot)$ operates locally on the target agent's data. In contrast, $\widetilde{\mathcal{A}}(\cdot)$ and $\widetilde{\mathcal{A}}_{\mathrm{dr}}(\cdot)$ collaboratively train models across source agents using FedAvg, exchanging only model updates–not raw data. Finally, $\mathcal{A}_{\mathrm{agg}}(\cdot)$ ensures privacy by relying solely on pre-computed estimates of conditional distributions and density ratios, requiring no access to raw data during aggregation.

**Remark 2.2.** *The estimation of $\widehat{F}_{\mathrm{mix}}(s \mid x)$ can be performed using either all available source agents or a selected subset of agents that are most relevant to the target domain. The criteria and implementation details for source agent selection are discussed in Section A.3 of the Appendix.*

### 2.3 Aggregate Conditional Distribution Estimators

This section formalizes a general framework for obtaining the aggregate conditional distribution estimator $\widehat{F}_{\mathrm{agg}}(s \mid x)$ given $\widehat{F}_{\mathrm{mix}}(s \mid x)$, $\widehat{r}(x, s)$ and $\widehat{F}(s \mid x)$. Without loss of generality, we assume $f(s \mid x) = 0$ for all $s < 0$. We further define $\widehat{r}(x) = \mathbb{E}_{S \sim \widehat{F}_{\mathrm{mix}}(s \mid x)} \widehat{r}(x, S)$ and $\widehat{r}(s \mid x) = \widehat{r}(x, s)/\widehat{r}(x)$. Then, both $\mathbb{E}_{S \sim \widehat{F}(s \mid x)} \mathbb{1}(S \le s)$ and $\mathbb{E}_{S \sim \widehat{F}_{\mathrm{mix}}(s \mid x)}\{\mathbb{1}(S \le s)\widehat{r}(S \mid x)\}$ can serve as estimators of $F(s \mid x)$. Introducing a weight function $\omega(x)$, we aggregate these estimators through

the following combination:

$$\widehat{F}_{\text{agg}}^{\omega}(s \mid x) = \{1 + \omega(x)\}^{-1}\Big[\mathbb{E}_{S\sim\widehat{F}(s|x)}\mathbb{1}(S \le s) + \omega(x)\mathbb{E}_{S\sim\widehat{F}_{\text{mix}}(s|x)}\{\mathbb{1}(S \le s)\widehat{r}(S \mid x)\}\Big].$$

In our framework, only $\widehat{r}(x,s)$ is available, and computing $\widehat{r}(s \mid x)$ requires estimating $r(x)$, potentially introducing additional error. To avoid this, we adopt the simplification $\omega(x) = \widehat{r}(x)$, leading to the following estimator:

$$\widehat{F}_{\text{agg}}(s \mid x) = \{1 + \widehat{r}(x)\}^{-1}\Big[\mathbb{E}_{S\sim\widehat{F}(s|x)}\mathbb{1}(S \le s) + \mathbb{E}_{S\sim\widehat{F}_{\text{mix}}(s|x)}\{\mathbb{1}(S \le s)\widehat{r}(x,S)\}\Big]. \quad (5)$$

In practice, the scalar term $1/\{1 + \widehat{r}(x)\}$ need not be estimated explicitly, as the estimator is obtained by weighting samples from the two distributions and computing the proportion below the threshold $s$. When integrated with the aggregation scheme based on engression [35] developed in Section A.1.3 of the Appendix, the estimator $\widehat{F}_{\text{agg}}(s \mid x)$ in (5) admits a simple form.

**Remark 2.3.** *Other choices of $\omega(x)$ are possible, but setting $\omega(x) = \widehat{r}(x)$ circumvents the need to explicitly estimate $r(x)$ and has been found to perform well in numerical experiments.*

## 2.4 Theoretical Analysis

In this section, we establish marginal coverage of our proposed PFCP and its test-conditional coverage property compared to GLCP. Let $V_{\text{agg},i} = \widehat{F}_{\text{agg}}(s(X_i, Y_i) \mid X_i)$ for $i \in [n+1]$, where $V_{\text{agg},i} = V_{\text{agg},i}^y$ for $i \in [n]$ and $V_{\text{agg},n+1} = V_{\text{agg},n+1}^{Y_{n+1}}$. The next theorem offers the marginal coverage guarantee.

**Theorem 2.4** (Marginal Validity)**.** *Assume $(X_1, Y_1), \ldots, (X_n, Y_n), (X_{n+1}, Y_{n+1})$ are i.i.d. from $P$, and $\widehat{F}_{\text{agg}}(s \mid x)$ is pre-trained using training data, which is independent of the calibration dataset. Then $\mathbb{P}(Y_{n+1} \in \widehat{C}_{\alpha}^{\text{PFCP}}(X_{n+1})) \ge 1 - \alpha$. In addition, if ties between $V_{\text{agg},1}, \ldots, V_{\text{agg},n+1}$ occur with probability zero, then*

$$\mathbb{P}(Y_{n+1} \in \widehat{C}_{\alpha}^{\text{PFCP}}(X_{n+1})) \in \left[1 - \alpha, 1 - \alpha + (n+1)^{-1}\right).$$

**Remark 2.5.** *Theorem 2.4 also holds if $\widehat{F}_{\text{agg}}(s \mid x)$ relies on calibration and test data but is invariant to the permutation of the combination of calibration and test data. However, in personalized federated setting, this will incur great data communication.*

Define $\delta_1(x; \widehat{F}) = d_{\text{TV}}(\widehat{F}(\cdot \mid x), F(\cdot \mid x))$, where $d_{\text{TV}}$ denotes the total variation. The next lemma provides an upper bound on the test-conditional miscoverage error of GLCP.

**Lemma 2.6.** *Under the conditions of Theorem 2.4, let $F_V$ be the CDF of $\widehat{F}(s(X_1, Y_1) \mid X_1)$, which is supported on $[0, 1]$ and assumed to be continuous. Let $q = \lceil(1-\alpha)(n+1)\rceil/n$. Then, for any $\delta \in (0, 1)$,*

$$\left|\mathbb{P}\big(Y_{n+1} \notin \widehat{C}_{\alpha}^{\text{GLCP}}(X_{n+1}) \mid X_{n+1} = x\big) - \alpha\right|$$
$$\le \max\left\{|1 - \alpha - F_V^{-1}(q - \epsilon) + \delta|, |1 - \alpha - F_V^{-1}(q + \epsilon + n^{-1}) - \delta|\right\} + \delta_1(x; \widehat{F}),$$

*where $\epsilon = \{(2n)^{-1}\ln(2/\delta)\}^{1/2}$.*

The result of $\widehat{C}_{\alpha}^{\text{PFCP}}(X_{n+1})$ can be derived by replacing $\widehat{F}(s \mid x)$ with $\widehat{F}_{\text{agg}}(s \mid x)$, and $F_V$ with the CDF of $\widehat{F}_{\text{agg}}(s(X_1, Y_1) \mid X_1)$. While the first term captures the global calibration error, the second term quantifies localized distributional estimation accuracy. The superiority of $\widehat{C}_{\alpha}^{\text{PFCP}}(X_{n+1})$ lies in its reduction of $\delta_1(x; \widehat{F}_{\text{agg}})$ compared to $\delta_1(x; \widehat{F})$, achieved through its federated learning architecture that uses $\widehat{r}(x,s)$. This comparison will be formally established in Theorem 2.8.

**Assumption 1.** For each $x \in \mathcal{X}$, the score $s(x, Y_1)$ on the target agent is bounded in $[0, M(x)]$, and $f(s \mid x) \ge 1/L_f(x)$ for $s \in (0, M(x))$, where $M(x) < \infty$ and $0 < L_f(x) < \infty$.

**Assumption 2.** For each $x \in \mathcal{X}$, there exist constants $0 < L_0(x), L_K(x) < \infty$, such that: (i) $\inf_{k\in[K]} \pi_k f_k(x)/\{\sum_{l=1}^{K} \pi_l f_l(x)\} \ge 1/L_K(x)$; (ii) for any $s$, there exists $k \in [K]$ satisfies $f(s \mid x)/f_k(s \mid x) \le L_0(x)$.

**Assumption 3.** The support of $s$ under $\widehat{r}(x,s)$ is bounded in $[0, M_1(x)]$. For any $x \in \mathcal{X}$, there exists $s \in [0, M_1(x)]$ such that $\widehat{r}(x,s) > 0$, and $\widehat{r}(x,s)$ is Lipschitz continuous in $s$ on its support with constant $L(\widehat{r}, x)$.

**Remark 2.7.** *If the distribution of $Y_1$ conditional on $X_1 = x$ is bounded with density bounded away from zero, and the conformity score function is $s(x, y) = |y - \widehat{\mu}(x)|$, where the pretrained predictor $\widehat{\mu}(x)$ estimates $\mathbb{E}(Y_1 \mid X_1 = x)$ and lies within the support of $Y_1$, then Assumption 1 holds. Assumption 2 has two key implications. First, each source's covariate distribution contributes nontrivially to the overall mixture (non-negligible mixture weight). Second, the scores for a given $x$ in the target agent are also observable in at least one source agent. Intuitively, this requires the source agents to possess richer or more informative data than the target agent, or more precisely, what we truly require is that the mixture of source distributions contains sufficient information about the target, meaning the $\sum_{k=1}^{K} \pi_k f_k(s \mid x)/f(s \mid x)$ has a positive lower bound, which is a necessary condition for the source data to meaningfully assist the target. Finally, Assumption 3, which requires the estimated density ratio $\widehat{r}(x, s)$ to have bounded support and be Lipschitz continuous, can be enforced in practice by truncating extreme score and ratio values, a straightforward step when employing continuous neural networks or predictors with bounded parameters.*

Define $\widetilde{\delta}(x; \widehat{r}) = \int_0^\infty f_{\text{mix}}(s \mid x)|\widehat{r}(x, s) - r(x, s)|ds$, and $\delta_2(x; \widehat{F}_{\text{mix}})$ as the $L_2$-distance between the density of $\widehat{F}_{\text{mix}}(\cdot \mid x)$ and $F_{\text{mix}}(\cdot \mid x)$, and $L_2(x; \widehat{r}) = \{\int_0^\infty \widehat{r}^2(s \mid x)ds\}^{-1/2}$.

**Theorem 2.8.** *Under the conditions of Theorem 2.4, Assumptions 1–3, we have $L_2(x; \widehat{r}) < \infty$ and $\widehat{F}_{\text{agg}}(s \mid x)$ in Section 2.3 satisfies that $2\delta_1(x; \widehat{F}_{\text{agg}})$ is bounded by*

$$2\delta_1(x; \widehat{F}_{\text{agg}}) \leq \{1 + \widehat{r}(x)\}^{-1} \left[2\delta_1(x; \widehat{F}) + \widetilde{\delta}(x; \widehat{r}) + |\widehat{r}(x) - r(x)| + L_2(x; \widehat{r})\delta_2(x; \widehat{F}_{\text{mix}})\right].$$

Theorem 2.8 implies that $\delta_1(x; \widehat{F}_{\text{agg}}) < \delta_1(x; \widehat{F})$ holds when $L_2(x; \widehat{r})\delta_2(x; \widehat{F}_{\text{mix}}) + \widetilde{\delta}(x; \widehat{r}) + |\widehat{r}(x) - r(x)| < 2\widehat{r}(x)\delta_1(x; \widehat{F})$. In practice, this condition is attainable under mild settings. With access to sufficient data from source agents, the estimation error $\delta_2(x; \widehat{F}_{\text{mix}})$ becomes asymptotically negligible. Moreover, density ratio estimation can be framed as a binary classification task, which is often easier to learn than conditional distribution estimation under similar conditions. Taken together, these observations suggest that $\widehat{F}_{\text{agg}}(s \mid x)$ is likely to yield improved estimation of $F(s \mid x)$ over $\widehat{F}(s \mid x)$, and thus PFCP is expected to outperform GLCP.

On the other hand, if the density ratio is poorly estimated, the federated conditional distribution estimator may not be more accurate, i.e. $\delta_1(x; \widehat{F}_{\text{agg}}) \geq \delta_1(x; \widehat{F})$, then the aggregation may not result in any improvement. Furthermore, if the source distribution significantly differs from the target distribution, for instance, when the support of $P$ and any $P_k$ do not intersect, the information from source agents cannot help improve the conformal set on the target agent. Additionally, when the data on the target agent is extremely limited, insufficient to yield a reliable estimate of the conditional distribution $F(s \mid x)$, both GLCP and PFCP may fail to deliver conformal sets with satisfying test-conditional coverage, though PFCP may still outperform GLCP in this regard.

## 3 Experiments

We conduct experiments on synthetic data and five real datasets to demonstrate marginal coverage validity of our proposed PFCP compared to FedCP [23], FedCP-QQ [16], CPlab [27] and CPhet [28] and its local coverage improvement relative to GLCP. The nominal coverage level is set to $1 - \alpha = 90\%$. For a prediction set $\widehat{C}_\alpha(X_{n+1})$, the following criteria are used for comparison: (1) marginal coverage $\mathbb{P}(Y_{n+1} \in \widehat{C}_\alpha(X_{n+1}))$; (2) test-conditional miscoverage error $\mathbb{E}\{|\mathbb{P}(Y_{n+1} \in \widehat{C}_\alpha(X_{n+1}) \mid X_{n+1}) - (1 - \alpha)|\}$; and (3) average size of the prediction set. All simulation results are based on 100 replications. The codes are available in the repository `https://github.com/OswinMin/PFCP`.

### 3.1 Synthetic Data

In our synthetic experiments, the regression model for target agent data is $Y_i = \mu(X_i) + \epsilon(X_i)$, where $\mu(x) = \mathbb{E}(Y_i \mid X_i = x)$ is the conditional mean function and the residual $\epsilon(X_i)$ may depend on $X_i$. For agent $k \in [K]$, the model is $Y_{k,i} = \mu_k(X_{k,i}) + \epsilon_k(X_{k,i})$. We set $\mu(x) = \sum_{i=1}^{d} x_i$ and $\mu_k(x) = 2/3 \sum_{i=1}^{d} x_i$ for $x = (x_1, \ldots, x_d)^\top \in \mathbb{R}^d$. The covariates $X_i$ and $X_{k,i}$ are sampled from the $d$-dimensional standard normal distribution $N_d(\mathbf{0}_d, \mathbf{I}_d)$. We consider two setting of heterogeneity across agents:

**S1:** $\epsilon(x) \sim N\big(0, \big| \sum_{i=1}^d |x_i|/\sqrt{d} - \sqrt{2d/\pi} \big|^2\big)$, $\epsilon_k(x) \sim N\big(0, \gamma_k \big| \sum_{i=1}^d |x_i|/\sqrt{d} - \sqrt{2d/\pi} \big|^2\big)$.

**S2:** $\epsilon(x) \sim N\big(0, \big| \cos\big(\sum_{i=1}^d x_i\big) \big|^2\big)$, $\epsilon_k(x) \sim N\big(0, \gamma_k \big| \cos\big(\sum_{i=1}^d x_i\big) \big|^2\big)$.

Here, $\gamma_k$ controls the level of heterogeneity across agents. The target agent dataset is divided into four parts: a predictor training set (I), a conditional distribution training set (II), a calibration dataset (III), and a test dataset (IV), where the first three datasets share the same data volume $n$. To ensure each agent has the same volume of available data, the source agent data set is divided into two parts (I) and (II), each with a data volume $3n/2$.

We set $n = 100$ in this experiment. The pretrained predictors $\widehat{\mu}(\cdot)$ and $\{\widehat{\mu}_k(\cdot)\}_{k=1}^K$ are obtained using neural networks with same hidden layers $30 \to 30$. The score function is defined as the residue $s(x, y) = |y - \widehat{\mu}(x)|$ and $s_k(x, y) = |y - \widehat{\mu}_k(x)|$. We report the experimental results evaluating the performance of PFCP across different covariate dimensions, with $d \in \{10, 15, 20, 25, 30\}$. Additional simulations exploring different source agent heterogeneity, quantities of source agents, and required coverage level and source agent selection methods can be found in Section A.4 in the Appendix. We consider two different sampling schemes for generating $\gamma_k$, applied under settings **S1** and **S2**. Specifically, in scenarios **S1.1** and **S2.1**, we sample $\gamma_k \sim U[0.8, 1]$, while in **S1.2** and **S2.2**, we use $\gamma_k \sim U[1, 1.4]$. We compare two density ratio estimation methods within the PFCP framework. **PFCP1** uses logistic regression, while **PFCP2** employs a neural network with a 30-unit hidden layer, followed by Platt calibration. In case **PFCP2**, the training data are randomly split in a 9:1 ratio for classifier training and Platt calibration.

**Results.** The top row of Figure 1 presents the marginal coverage rates of all methods across four configurations. It shows that both GLCP and PFCP achieve valid marginal coverage, whereas FedCP and FedCP-QQ fail to meet this guarantee. Consequently, the bottom row of Figure 1 focuses exclusively on GLCP and PFCP for test-conditional miscoverage error comparison. The results demonstrate that PFCP consistently outperforms GLCP in terms of conditional coverage, with the performance gap getting larger as the covariate dimension $d$ increases, highlighting the robustness of PFCP in high-dimensional settings. Additionally, PFCP2, which employs a calibrated classifier via Platt scaling after neural feature extraction, generally exhibits slightly lower test-conditional miscoverage error than PFCP1, which uses the logistic regression-based density ratio estimation method, especially in high-dimensional settings.

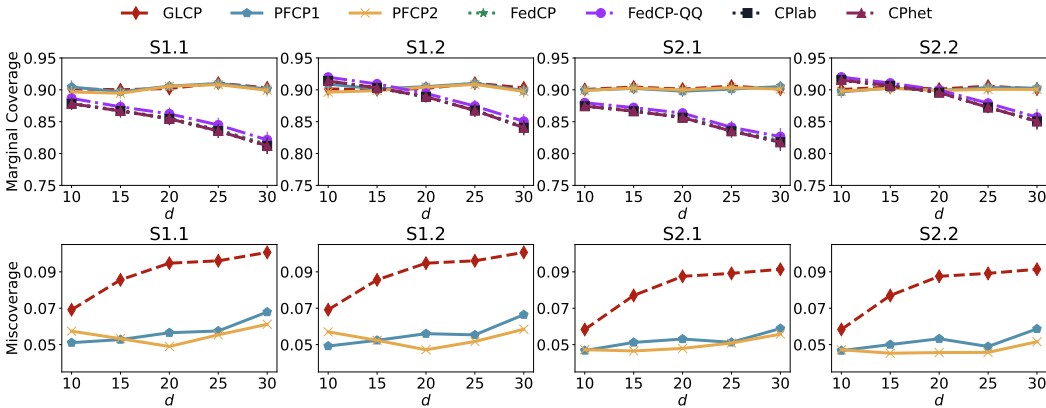

Figure 1: Marginal coverage (top row) and test-conditional miscoverage error (bottom row) under scenarios **S1.1**, **S1.2**, **S2.1**, and **S2.2**.

## 3.2 Real data analysis

We evaluate our proposed PFCP method on five public-domain regression datasets also considered by [32, 33, 16]: physicochemical properties of protein tertiary structure (BIO) [30], bike sharing (BIKE) [8], communities and crimes (CRIME) [31], Tennessee's student teacher achievement ratio (STAR) [1], concrete compressive strength (CONCRETE) [43], and a derma image classification dataset (DERMA) [42].

We first partition each dataset into multiple agents, with the partitioning strategy detailed in Section A.5 in the Appendix. We set $n = 50$ for the CONCRETE dataset and $n = 100$ for other four datasets. In each repeated experiment, the number of source agents (excluding the target) varies across benchmark datasets: 20 for BIO, 12 for BIKE, 4 for CRIME, 8 for STAR, 5 for CONCRETE and 4 for DERMA. To ensure fair comparison, all agents use neural networks with identical architectures for point estimation, with hidden layers of size $30 \rightarrow 30$, and the engressor [35] is trained using the same network structure with hidden layers of $100 \rightarrow 100$, all trained for the same number of epochs and learning rate. For regression tasks, the score function still uses the residue score. For classification problem, the point estimate $\widehat{\mu}$ and $\widehat{\mu}_k$ output a probability vector, and the score is denoted as $s(x, y) = 1 - \{\widehat{\mu}(x)\}_y, s_k(x, y) = 1 - \{\widehat{\mu}_k(x)\}_y$, where the subscript indicates the $y$-th element of the probability vector.

**Results**: The Marginal Coverage part of Table1 summarizes the marginal coverage rates of four methods across 100 repeats. Both GLCP and PFCP achieve reliable marginal coverage close to the required $1 - \alpha = 90\%$ level, while FedCP, FedCP-QQ, CPlab and CPhet deviate significantly from the target in most scenarios. This highlights substantial data heterogeneity across agents in real-world settings, where achieving nominal marginal coverage relies on rare coincidental alignment–a high-risk strategy lacking robustness. Results violating marginal validity are boldfaced and marked with $\times$. The conditional miscoverage error requires the calculation of conditional coverage rates, with the computation method detailed in Section A.5 in the Appendix. Additionally, the Miscoverage and Size part of Table1 reports the conditional miscoverage errors and mean set size of GLCP and PFCP. Across all scenarios, PFCP consistently outperforms GLCP, with improvement percentages shown in parentheses.

Table 1: Marginal Coverage, miscoverage rate, and size of prediction sets.

|  | Dataset | GLCP | PFCP | FedCP | FedCP-QQ | CPlab | CPhet |
|---|---|---|---|---|---|---|---|
| Marginal | BIO | 0.902 | 0.903 | **0.993**$\times$ | **0.970**$\times$ | **0.978**$\times$ | **0.981**$\times$ |
|  | BIKE | 0.900 | 0.900 | **0.885**$\times$ | 0.890 | **0.885**$\times$ | **0.883**$\times$ |
|  | CRIME | 0.900 | 0.896 | **0.866**$\times$ | **0.852**$\times$ | **0.862**$\times$ | **0.864**$\times$ |
|  | STAR | 0.898 | 0.899 | 0.897 | 0.892 | 0.897 | 0.898 |
|  | CONCRETE | 0.903 | 0.905 | **0.947**$\times$ | **0.963**$\times$ | **0.949**$\times$ | **0.946**$\times$ |
|  | DERMA | 0.895 | 0.899 | **0.824**$\times$ | **0.809**$\times$ | **0.880**$\times$ | **0.868**$\times$ |
| Miscoverage | BIO | 0.0315 | **0.0199** | 0.0941 | 0.0753 | 0.0822 | 0.0844 |
|  | BIKE | 0.0234 | **0.0193** | 0.0678 | 0.0647 | 0.0681 | 0.0690 |
|  | CRIME | 0.0387 | **0.0268** | 0.0426 | 0.0495 | 0.0439 | 0.0429 |
|  | STAR | 0.0392 | **0.0244** | 0.0502 | 0.0507 | 0.0498 | 0.0493 |
|  | CONCRETE | 0.0366 | **0.0238** | 0.0582 | 0.0675 | 0.0600 | 0.0580 |
|  | DERMA | 0.0300 | **0.0230** | 0.0884 | 0.0976 | 0.0616 | 0.0659 |
| Size | BIO | 0.5144 | **0.5032** | 0.9991 | 0.7087 | 0.8054 | 0.8271 |
|  | BIKE | 4.0968 | **3.9645** | 4.0044 | 4.0910 | 3.9959 | 3.9671 |
|  | CRIME | 5.1830 | **4.4961** | 3.9176 | 3.7724 | 3.8802 | 3.8855 |
|  | STAR | 48.4987 | 43.2931 | 42.7556 | **42.1185** | 42.8081 | 43.0090 |
|  | CONCRETE | 34.4859 | **28.3797** | 34.7115 | 38.9746 | 35.1545 | 34.6905 |
|  | DERMA | 2.4926 | **2.4430** | 1.2490 | 1.1943 | 1.5649 | 1.4810 |

To intuitively demonstrate PFCP's improvement in conditional coverage over GLCP, Figure 2 visualizes the conditional miscoverage errors across subspaces for the CONCRETE dataset (results for other datasets are in the Appendix, see Figure 12 to 16). The figure shows that GLCP exhibits highly heterogeneous coverage across subspaces, while PFCP achieves more stable and better controlled miscoverage rates closer to the target $\alpha$.

To further investigate how the number of source agents impacts target agent performance, we hypothesize that increasing the number of source agents with distributional alignment to the target should enhance task performance, while using an optimal subset of source agents becomes critical when distribution shifts are significant. Figure 3 demonstrates this phenomenon on the CRIME dataset with $\alpha = 0.1$ (with $n = 75$ adjusted to accommodate more source agents), where target performance improves steadily as the number of source agents grows. Results for other datasets are provided in the Appendix (see Figure 17 to 21), reinforcing that task-specific distributional alignment–rather than simply maximizing source agents–drives reliable performance gains.

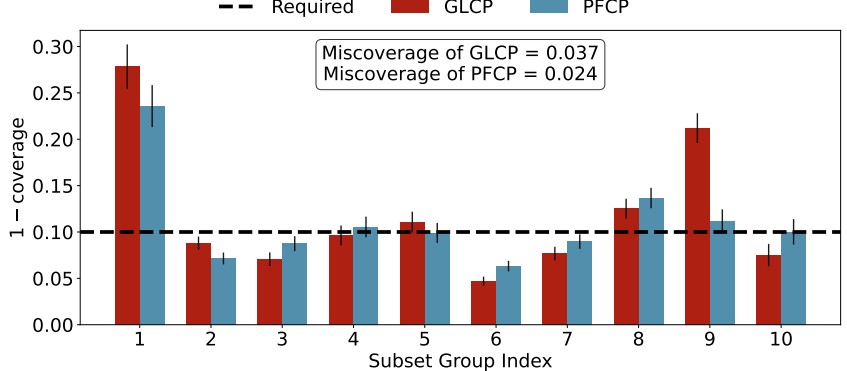

Figure 2: Conditional coverage over all subspace for CONCRETE dataset.

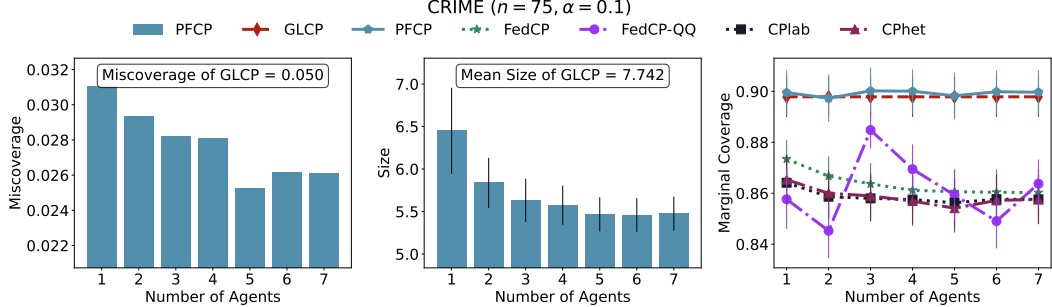

Figure 3: Impact of source agent quantity on target task performance for CRIME dataset.

# 4 Conclusion

We present Personalized Federated Conformal Prediction (PFCP), a novel framework that addresses both agent-specific and instance-specific uncertainty quantification in federated learning. By integrating localized conformal prediction with personalized federated models, PFCP guarantees marginal coverage for target agents while significantly improving test-conditional coverage through privacy-preserving knowledge transfer from source agents. Theoretical analysis and empirical results demonstrate that PFCP achieves tighter prediction sets and better conditional reliability compared to methods relying solely on local data. This work bridges the critical gap between federated learning's privacy constraints and risk-sensitive applications' demand for instance-aware uncertainty quantification. The framework's robustness to distribution shifts and its compatibility with arbitrary data types position PFCP as a principled solution for safety-critical federated systems requiring personalized reliability guarantees.

## Acknowledgments and Disclosure of Funding

Peng was supported by ARC (Grant No. LP240100101). Zou was supported by the National Key R&D Program of China (Grant No. 2022YFA1003703) and the National Natural Science Foundation of China (Grant Nos. 12231011 and 12531011).

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

# A  Technical Appendices and Supplementary Material

## A.1  Implementation Details

### A.1.1  Engression as Conditional Distribution Estimator

Engression consider the problem of estimating the conditional distribution $Y \mid X = x$ via a generative approach. Let $X \in \mathcal{X} \subseteq \mathbb{R}^d$, $Y \in \mathcal{Y} \subseteq \mathbb{R}$ and $\varepsilon \in \mathcal{E} \subseteq \mathbb{R}^{d'}$ be the input, output variables and random noise, respectively. Engression constructs a generator $G(x, \varepsilon; \theta) : \mathcal{X} \times \mathcal{E} \to \mathcal{Y}$ that maps a covariate $x$ and a random noise variable $\varepsilon \sim P_\varepsilon$ to a sample from the conditional distribution $P(Y|X = x)$, where $P_\varepsilon$ is a pre-defined distribution and $\theta \in \Theta$ is the parameter space. The randomness of $\varepsilon$ enables the generator to produce diverse outputs for a fixed $x$, thereby approximating the full conditional distribution [35].

To evaluate the quality of the generated distribution, engression uses the energy score, a strictly proper scoring rule. For a distribution $P$ and an observation $y$, the energy score is defined as

$$\text{ES}(P, y) = \frac{1}{2}\mathbb{E}_{Z, Z' \sim P}\|Z - Z'\| - \mathbb{E}_{Z \sim P}\|Z - y\|,$$

where $\|\cdot\|$ denotes the Euclidean norm. A scoring rule $S(P, y)$ is strictly proper if it satisfies

$$\mathbb{E}_{Y \sim P_0}\left[S(P_0, Y)\right] < \mathbb{E}_{Y \sim P_0}\left[S(Q, Y)\right],$$

for any candidate distribution $Q \neq P_0$, where $P_0$ is the true data-generating distribution. This property ensures that the oracle objective $\min_Q \mathbb{E}_{(X,Y)}\left[S(Q, Y)\right]$ is uniquely minimized at $Q = P_0$. The energy score is a strictly proper scoring rule that can be employed to identify the true underlying distribution.

**Local Training Procedure** $\mathcal{A}(\cdot)$**:** Given $n$ observed data pairs $\{(X_i, Y_i)\}_{i=1}^n$, engression defines an empirical loss function to approximate the oracle objective. For each $X_i$, we generate $m$ independent noise samples $\{\varepsilon_{i,j}\}_{j=1}^m$ with $\varepsilon_{i,j} \sim P_\varepsilon$, and compute the loss $\mathcal{L}^{\text{Eng}}(\theta)$ as

$$
\mathcal{L}^{\text{Eng}}(\theta)
$$
$$
= \frac{1}{n}\sum_{i=1}^n \left\{ \frac{1}{m}\sum_{j=1}^m \|Y_i - G(X_i, \varepsilon_{i,j}; \theta)\| - \frac{1}{2m(m-1)}\sum_{j \neq j'} \|G(X_i, \varepsilon_{i,j}; \theta) - G(X_i, \varepsilon_{i,j'}; \theta)\| \right\}.
$$
$$(6)$$

The first term measures the alignment between generated samples and the observed data, while the second term regularizes the dispersion of the generated distribution by encouraging pairwise differences between samples. Minimizing this loss jointly optimizes both fidelity and diversity of the estimated conditional distribution. The pseudocode of $\mathcal{A}(\cdot)$ is provided in Algorithm 1.

---

**Algorithm 1** Local Training Procedure of Engression

---

**Input:** Data pairs $\{(X_i, Y_i)\}_{i=1}^n$, initial parameter $\widehat{\theta}_0$, training steps $T$, learning rate $\zeta_0$, noise distribution $P_\epsilon$, number of noise samples $m$

  **For** $t = 1$ **to** $T$ **do**

1:  Generate $m$ i.i.d. samples $\{\varepsilon_{t,i,j}\}_{j=1}^m$ with $\varepsilon_{t,i,j} \sim P_\varepsilon$ for each $i \in [n]$

2:  Replace $\varepsilon_{i,j}$ by $\varepsilon_{t,i,j}$ in (6) of $\mathcal{L}^{\text{Eng}}(\widehat{\theta}_{t-1})$, and take gradient of $\theta$ at $\widehat{\theta}_{t-1}$ as $\nabla_\theta \mathcal{L}^{\text{Eng}}(\widehat{\theta}_{t-1})$

3:  Update parameters by $\widehat{\theta}_t = \widehat{\theta}_{t-1} - \zeta_0 \nabla_\theta \mathcal{L}^{\text{Eng}}(\widehat{\theta}_{t-1})$

4: **return** $\widehat{\theta}_T$.

---

**Federated Training Procedure** $\widetilde{\mathcal{A}}(\cdot)$**:** We extend the engression algorithm to a federated version. Given $K$ groups of observed data pairs $\{(X_{k,i}, Y_{k,i})\}_{i=1}^{n_k}$, $k = 1, \ldots, K$, generate independent noise samples $\{\varepsilon_{k,i,j}\}_{1 \leq k \leq K, 1 \leq i \leq n_k}$ with $\varepsilon_{i,j} \sim P_\varepsilon$. Define $\mathcal{L}_k^{\text{Eng}}(\theta)$ as the engression loss function for agent $k$, obtained by replacing all instances of $X_i, Y_i, \varepsilon_{i,j}$ with $X_{k,i}, Y_{k,i}, \varepsilon_{k,i,j}$, and the sample size

$n$ with $n_k$ in $\mathcal{L}^{\mathrm{Eng}}(\theta)$:

$$\mathcal{L}_k^{\mathrm{Eng}}(\theta) = \frac{1}{n_k}\sum_{i=1}^{n_k}\left\{\frac{1}{m}\sum_{j=1}^{m}\|Y_{k,i} - G(X_{k,i}, \varepsilon_{k,i,j}; \theta)\| - \right.$$

$$\left. \frac{1}{2m(m-1)}\sum_{j\neq j'}\|G(X_{k,i}, \varepsilon_{k,i,j}; \theta) - G(X_{k,i}, \varepsilon_{k,i,j'}; \theta)\|\right\}. \tag{7}$$

Thus the federated empirical loss can be represented as

$$\widetilde{\mathcal{L}}^{\mathrm{Eng}}(\theta) = \left(\sum_{k=1}^{K} n_k\right)^{-1}\sum_{k=1}^{K} n_k \mathcal{L}_k^{\mathrm{Eng}}(\theta).$$

To minimize loss $\widetilde{\mathcal{L}}^{\mathrm{Eng}}(\theta)$, each agent computes the average gradient over its local dataset and participates in the aggregation of $K$ gradients to iteratively update the model $G(\cdot, \cdot; \theta)$ [24]. The parameter updating procedure is similar to that in Section A.1.2. The pseudocode of $\widetilde{\mathcal{A}}(\cdot)$ is provided in Algorithm 2.

---

**Algorithm 2** Federated Training Procedure of Engression

---

**Input:** Data pairs $\{(X_{k,i}, Y_{k,i})\}_{i=1}^{n_k}$, $k \in [K]$, initial parameter $\widehat{\theta}_0$, training steps $T$, learning rate $\zeta_0$, noise distribution $P_\epsilon$, number of noise samples $m$

    **For** $t = 1$ **to** $T$ **do**

        **For** $k = 1$ **to** $K$ **do**

1:        Generate $m$ i.i.d. samples $\{\varepsilon_{t,k,i,j}\}_{j=1}^m$ with $\varepsilon_{t,k,i,j} \sim P_\varepsilon$ for each $i \in [n_k]$

2:        Replace $\varepsilon_{k,i,j}$ by $\varepsilon_{t,k,i,j}$ in (7) of $\mathcal{L}_k^{\mathrm{Eng}}(\widehat{\theta}_{t-1})$, and take gradient of $\theta$ at $\widehat{\theta}_{t-1}$ as $\nabla_\theta \mathcal{L}_k^{\mathrm{Eng}}(\widehat{\theta}_{t-1})$

3:      Communicate and aggregate $K$ gradients with weights proportion to $n_1, \ldots, n_K$, the aggregated gradient is denoted as $(\sum_{k=1}^K n_k)^{-1}\sum_{k=1}^K n_k \nabla_\theta \mathcal{L}_k^{\mathrm{Eng}}(\widehat{\theta}_{t-1})$

4:      Update parameters by $\widehat{\theta}_t = \widehat{\theta}_{t-1} - \zeta_0(\sum_{k=1}^K n_k)^{-1}\sum_{k=1}^K n_k \nabla_\theta \mathcal{L}_k^{\mathrm{Eng}}(\widehat{\theta}_{t-1})$

5: **return** $\widehat{\theta}_T$.

---

### A.1.2 Federated Density Ratio Estimation

The density ratio $r(x, s) = f(x, s)/f_{\mathrm{mix}}(x, s)$, which quantifies the discrepancy between the target distribution $P$ and the mixture $\sum_{k=1}^K \pi_k P_k$, can be reduced to a binary classification problem [36]. Specifically, we first train a classifier to distinguish samples from the target versus mixture distributions, then apply calibration methods [44], i.e., Platt-calibration [29] to convert classifier outputs into well-calibrated probability estimates. This framework naturally extends to federated settings: the classification component is implemented through standard FedAvg optimization across agents, while Platt calibration–performed locally using only prediction scores–requires no exchange of private data.

We formulate a binary classification task where samples from the target distribution $(X, s(X, Y)) \sim f(x, s)$ are assigned label $l = 1$, while those from the mixture distribution $(X_{k,\cdot}, s_k(X_{k,\cdot}, Y_{k,\cdot})) \sim f_{\mathrm{mix}}(x, s)$ receive label $l = 0$. A global probabilistic classifier $\widetilde{f}(x, s; \gamma)$, where $\gamma \in \Gamma_{\mathrm{cl}}$ is the model parameter, implemented as a neural network, is trained collaboratively across all $K + 1$ agents (one target and $K$ auxiliary) using federated averaging [24]. During each communication round, local updates are computed by minimizing the cross-entropy loss:

$$\mathcal{L}_k^{\mathrm{cl}}(\gamma) = -n_k^{-1}\sum_{(x,y)\in\mathcal{D}_k}\log\left\{1 - \widetilde{f}(x, s_k(x, y); \gamma)\right\}, \tag{8}$$

$$\mathcal{L}_0^{\mathrm{cl}}(\gamma) = -n^{-1}\sum_{(x,y)\in\mathcal{D}_{\mathrm{tr}}}\log\widetilde{f}(x, s(x, y); \gamma). \tag{9}$$

The federated loss can be written as

$$\widetilde{\mathcal{L}}^{\mathrm{cl}}(\gamma) = \Big\{ \mathcal{L}_0^{\mathrm{cl}}(\gamma) + \Big( \sum_{k=1}^{K} n_k \Big)^{-1} \Big( \sum_{k=1}^{K} n_k \mathcal{L}_k^{\mathrm{cl}}(\gamma) \Big) \Big\}/2\,.$$

Model parameters are updated with gradient $\nabla_\gamma \widetilde{L}^{\mathrm{cl}}(\widehat{\gamma}_{t-1})$, where

$$\nabla_\gamma \widetilde{L}^{\mathrm{cl}}(\widehat{\gamma}_{t-1}) = \Big\{ \nabla_\gamma \mathcal{L}_0^{\mathrm{cl}}(\widehat{\gamma}_{t-1}) + \Big( \sum_{k=1}^{K} n_k \Big)^{-1} \Big( \sum_{k=1}^{K} n_k \nabla_\gamma \mathcal{L}_k^{\mathrm{cl}}(\widehat{\gamma}_{t-1}) \Big) \Big\}\,.$$

After $T$ training epochs, the final parameter is denoted by $\widehat{\gamma} = \widehat{\gamma}_T$.

To calibrate the classifier outputs into reliable density ratio estimates, we apply Platt scaling on a hold-out subset $\mathcal{D}_{\mathrm{hold}}$ from the target agent and other source agents. This involves fitting a logistic regression model to adjust the raw predictions by $\widehat{f}(x,s) = \sigma\big(a \cdot \widetilde{f}(x,s;\widehat{\gamma}) + b\big)$, where $\sigma(\cdot)$ is the sigmoid function and $(a,b)$ are calibration parameters optimized on $\mathcal{D}_{\mathrm{hold}}$. The density ratio is then estimated as $\widehat{r}(x,s) = \widehat{f}(x,s)/\big\{1 - \widehat{f}(x,s)\big\}$.

Data privacy is preserved by ensuring raw covariates $X$ and responses $Y$ never leave their originating agents — only logits $\widetilde{f}(X, s(X,Y); \widehat{\gamma})$ and model gradients are shared. This federated implementation adheres to data isolation constraints while mitigating distributional shifts through collaborative learning.

---

**Algorithm 3** Federated Density Ratio Estimation Procedure

---

**Input:** Datasets $\mathcal{D}_0 = \mathcal{D}_{\mathrm{tr}}$ and $\{\mathcal{D}_k\}_{k=1}^K$, initial parameter $\widehat{\gamma}_0$, training steps $T$, learning rate $\zeta_0$, pretrained score function $s(\cdot,\cdot)$ and $\{s_k(\cdot,\cdot)\}_{k=1}^K$, hold-out ratio $\eta$

1: Compute conformity scores $s(x,y)$ for all $(x,y) \in \mathcal{D}_0$
2: Split dataset $\mathcal{D}_0 = \mathcal{D}_{0,1} \cup \mathcal{D}_{0,2}$, where $\mathcal{D}_{0,1} \cap \mathcal{D}_{0,2} = \emptyset$ and $|\mathcal{D}_{0,2}| = \lceil \eta |\mathcal{D}_0| \rceil$
   **For $k = 1$ to $K$ do**
3:     Compute conformity scores $s_k(x,y)$ for all $(x,y) \in \mathcal{D}_k$
4:     Split dataset $\mathcal{D}_k = \mathcal{D}_{k,1} \cup \mathcal{D}_{k,2}$, where $\mathcal{D}_{k,1} \cap \mathcal{D}_{k,2} = \emptyset$ and $|\mathcal{D}_{k,2}| = \lceil \eta |\mathcal{D}_k| \rceil$
   **For $t = 1$ to $T$ do**
       **For $k = 1$ to $K$ do**
5:         Calculate the gradients of $\mathcal{L}_k^{\mathrm{cl}}(\gamma)$ with respect to $\gamma$ at $\widehat{\gamma}_{t-1}$ and denote the gradients as $\nabla_\gamma \mathcal{L}_k^{\mathrm{cl}}(\widehat{\gamma}_{t-1})$ using dataset $\mathcal{D}_{k,1}$.
6:         Communicate and aggregate $K+1$ gradients, the aggregated gradient is denoted as $\nabla_\gamma \widetilde{L}^{\mathrm{cl}}(\widehat{\gamma}_{t-1}) = \{\nabla_\gamma \mathcal{L}_0^{\mathrm{cl}}(\widehat{\gamma}_{t-1}) + (\sum_{k=1}^K n_k)^{-1}(\sum_{k=1}^K n_k \nabla_\gamma \mathcal{L}_k^{\mathrm{cl}}(\widehat{\gamma}_{t-1}))\}$
7:         Update parameters by $\widehat{\gamma}_t = \widehat{\gamma}_{t-1} - \nabla_\gamma \widetilde{L}^{\mathrm{cl}}(\widehat{\gamma}_{t-1})$.
8: Construct raw predictor $\widetilde{f}(x,s;\widehat{\gamma}_T)$
9: Locally calculate and construct logits data pairs $(\widetilde{f}(x, s(x,y);\widehat{\gamma}_T), 1)$ for $(x,y) \in \mathcal{D}_{0,2}$
   **For $k = 1$ to $K$ do**
10:     Locally calculate $(\widetilde{f}(x, s_k(x,y);\widehat{\gamma}_T), 0)$ for $(x,y) \in \mathcal{D}_{k,2}$
11: Communicate the $K+1$ logits data pairs and fit logistic regression $\widehat{f}(x,s) = \sigma\big(a \cdot \widetilde{f}(x,s;\widehat{\gamma}) + b\big)$
12: **return** Density ratio estimation $\widehat{r}(x,s) = \widehat{f}(x,s)/\big\{1 - \widehat{f}(x,s)\big\}$

---

### A.1.3 Aggregate Engression-Based Conditional Distribution Estimators

Let $G(x,\varepsilon;\widehat{\theta}_0)$ and $G(x,\varepsilon;\widehat{\theta}_K)$ denote the empirical estimators of the conditional distributions corresponding to densities $f(x,s)$ and $f_{\mathrm{mix}}(x,s)$, respectively. The true underlying parameters $\theta_0$ and $\theta_K$ are defined such that, for any fixed $x$, the random variables $G(x,\varepsilon;\theta_0)$ and $G(x,\varepsilon;\theta_K)$ follow distributions with conditional densities $f(s \mid x)$ and $f_{\mathrm{mix}}(s \mid x)$ respectively. Given covariate $x$, reweight samples from $G(x,\varepsilon;\theta_K)$ with weight proportion to $r(x,s)$ and we get samples with density $f(s \mid x)$.

Let $\overline{\varepsilon}_{0,m_0} = (\varepsilon_{0,1}, \ldots, \varepsilon_{0,m_0}), \overline{\varepsilon}_{K,m_K} = (\varepsilon_{K,1}, \ldots, \varepsilon_{K,m_K})$ be sampled i.i.d. from $(P_\varepsilon)^{m_0}$ and $(P_\varepsilon)^{m_K}$. We propose estimating $\mathbb{P}(s(X,Y) \leq s \mid X = x)$ by aggregating $G(x,\varepsilon;\widehat{\theta}_0)$ and

$G(x, \varepsilon; \widehat{\theta}_K)$ with $\widehat{r}(\cdot, \cdot)$ by:

$$\beta_{\widehat{r}, \widehat{\theta}_0, \widehat{\theta}_K}(s; x, \overline{\varepsilon}) = C_K(x, \overline{\varepsilon}_{0,m}, \overline{\varepsilon}_{K,m}) \sum_{k=0,K} \sum_{j=1}^{m_k} \widehat{r}_k(x, G(x, \varepsilon_{k,j}; \widehat{\theta}_k)) \mathbb{1}(G(x, \varepsilon_{k,j}; \widehat{\theta}_k) \leq s),$$

(10)

where $C_K(x, \overline{\varepsilon}_{0,m}, \overline{\varepsilon}_{K,m}) = \left\{ \sum_{k=0,K} \sum_{j=1}^{m_k} \widehat{r}_k(x, G(x, \varepsilon_{k,j}; \widehat{\theta}_k)) \right\}^{-1}$, and $\widehat{r}_0 \equiv 1, \widehat{r}_K = \widehat{r}$.

Therefore, the conformal score on calibration can be written as $V_i^y = \beta_{\widehat{r}, \widehat{\theta}_0, \widehat{\theta}_K}(s(X_i, Y_i); X_i, \overline{\varepsilon})$, for $i = 1, \ldots, n + 1$ and $Y_{n+1} = y$.

---

**Algorithm 4** Conditional Distribution Estimator Aggregation Procedure

---

**Input:** Two conditional distribution estimators $G(x, \varepsilon; \widehat{\theta}_0)$ and $G(x, \varepsilon; \widehat{\theta}_K)$, density ratio estimator $\widehat{r}_K = \widehat{r}$, noise distribution $P_\epsilon$, number of noise samples $m_0, m_K$

1: Sample independent noise vectors $\overline{\varepsilon}_{0,m_0} = (\varepsilon_{0,1}, \ldots, \varepsilon_{0,m_0}), \overline{\varepsilon}_{K,m_K} = (\varepsilon_{K,1}, \ldots, \varepsilon_{K,m_K})$ from $(P_\varepsilon)^{m_0}$ and $(P_\varepsilon)^{m_K}$
2: Calculate $\beta_{\widehat{r}, \widehat{\theta}_0, \widehat{\theta}_K}(s; x, \overline{\varepsilon})$ in (10) as an aggregated estimator of $\mathbb{P}(s(X, Y) \leq s \mid X = x)$
3: **return** $\beta_{\widehat{r}, \widehat{\theta}_0, \widehat{\theta}_K}(s; x, \overline{\varepsilon})$

---

### A.1.4 PFCP Algorithm

We provide the algorithm that follows PFCP procedure in Algorithm 5.

---

**Algorithm 5** Personalized Federated Conformal Prediction (PFCP)

---

**Input:** Training and calibration datasets from target agent $\mathcal{D}_{\mathrm{tr}}, \mathcal{D}_{\mathrm{cal}}$, datasets from $K$ source agents $\mathcal{D}_1, \ldots, \mathcal{D}_K$, pretrained score functions $s(\cdot, \cdot)$ and $\{s_k(\cdot, \cdot)\}_{k=1}^K$, algorithms $\mathcal{A}, \widetilde{\mathcal{A}}, \widetilde{\mathcal{A}}_{\mathrm{dr}}$ and $\mathcal{A}_{\mathrm{agg}}$, test point $X_{n+1}$, nominal coverage rate $1 - \alpha$.

    **Preparing Step:**

1: Calculate conformity scores $\{S_i\}_{i=1}^n, \{S_i'\}_{i=1}^n$ and $\{S_{k,i}\}_{i=1}^{n_k}$ for $k \in [K]$.
2: Obtain datasets $\widetilde{\mathcal{D}}_{\mathrm{tr}} = \{(X_i', S_i')\}_{i=1}^n$ and $\widetilde{\mathcal{D}}_k = \{(X_{k,i}, S_{k,i})\}_{i=1}^{n_k}$ for $k \in [K]$.
3: Train $\widehat{F}(s \mid x) = \mathcal{A}(\widetilde{\mathcal{D}}_{\mathrm{tr}})$ and $\widehat{F}_{\mathrm{mix}}(s \mid x) = \widetilde{\mathcal{A}}(\widetilde{\mathcal{D}}_1, \ldots, \widetilde{\mathcal{D}}_K)$.
4: Train $\widehat{r}(x, s) = \widetilde{\mathcal{A}}_{\mathrm{dr}}(\widetilde{D}_{\mathrm{tr}}, \widetilde{D}_1, \ldots, \widetilde{D}_K)$.
    **Conformal Prediction Step:**
5: Obtain aggregate estimator $\widehat{F}_{\mathrm{agg}}(s \mid x) = \mathcal{A}_{\mathrm{agg}}(\widehat{F}(s \mid x), \widehat{F}_{\mathrm{mix}}(s \mid x), \widehat{r}(x, s))$.
6: Calculate transformed scores $V_i^y = \widehat{F}_{\mathrm{agg}}(S_i^y \mid X_i)$ for $i \in [n+1]$.
7: Construct PFCP set $\widehat{C}_\alpha^{\mathrm{PFCP}}(X_{n+1})$ using (4).
8: **return** $\widehat{C}_\alpha^{\mathrm{PFCP}}(X_{n+1})$.

---

## A.2 Proofs

### A.2.1 Proof of Theorem 2.4

For notation simplicity, we omit the superscript $Y_{n+1}$ of $V_i^{Y_{n+1}}$, that is, the transformed scores $V_i = \widehat{F}_{\mathrm{agg}}(s(X_i, Y_i) \mid X_i), i = 1, \ldots, n + 1$. Due to the i.i.d. data pairs $(X_1, Y_1), \ldots, (X_{n+1}, Y_{n+1})$ drawn from $P$, and their independence from $\widetilde{\mathcal{D}}_{\mathrm{tr}}$ and $\widetilde{\mathcal{D}}_k, k = 1, \ldots, K$, the transformed scores $\{V_i\}_{i=1}^{n+1}$ are exchangeable. Let $V_{(k,n)}$ denote the $k$-th smallest value in $\{V_i\}_{i=1}^n$. By definition, as long as $\lceil (1-\alpha)(n+1) \rceil < n + 1$, we have:

$$Q\left(1 - \alpha; (n+1)^{-1}\left(\sum_{i=1}^n \delta_{V_i} + \delta_1\right)\right) = V_{(\lceil (1-\alpha)(n+1) \rceil, n)}.$$

(11)

Therefore,

$$\mathbb{P}\left(V_{n+1} \leq V_{(\lceil (1-\alpha)(n+1)\rceil,n)}\right) \overset{(i)}{=} \mathbb{P}\left(V_{n+1} \leq V_{(\lceil (1-\alpha)(n+1)\rceil,n+1)}\right)$$

$$\overset{(ii)}{=} \frac{1}{n+1} \sum_{i=1}^{n+1} \mathbb{P}\left(V_i \leq V_{(\lceil (1-\alpha)(n+1)\rceil,n+1)}\right)$$

$$= \frac{1}{n+1} \mathbb{E}\left[\sum_{i=1}^{n+1} \mathbb{1}\left\{V_i \leq V_{(\lceil (1-\alpha)(n+1)\rceil,n+1)}\right\}\right]$$

$$\overset{(iii)}{=} \frac{\lceil (1-\alpha)(n+1)\rceil}{n+1}.$$

Here, (i) is because $V_{n+1} \leq V_{(k,n)}$ if and only if $V_{n+1} \leq V_{(k,n+1)}$. Indeed, if $V_{n+1} \leq V_{(k,n)}$, then $V_{(k,n+1)}$ is the larger of $V_{(k-1,n)}$ and $V_{n+1}$; therefore, $V_{(k,n+1)} \geq V_{n+1}$. Conversely, if $V_{n+1} \leq V_{(k,n+1)}$, then also $V_{n+1} \leq V_{(k,n)}$ because $V_{(k,n+1)} \leq V_{(k,n)}$. (ii) follows from the exchangeability of $\{V_i\}_{i=1}^{n+1}$, and (iii) follows from the no-tie assumption.

Then by the fact $1 - \alpha \leq \frac{\lceil (1-\alpha)(n+1)\rceil}{n+1} < \frac{(1-\alpha)(n+1)+1}{n+1} = 1 - \alpha + \frac{1}{n+1}$, we achieve the second result. If the no-tie assumption does not hold, the equality in (iii) can be replaced by an inequality "$\geq$" based on the definition of $V_{(k,n+1)}$. Consequently, we still obtain marginal validity of $\widehat{C}_\alpha^{\text{PFCP}}(X_{n+1})$.

### A.2.2 Proof of Lemma 2.6

Recall that

$$\widehat{C}_\alpha^{\text{GLCP}}(X_{n+1}) = \left\{y : V_{n+1}^y \leq Q\left(1 - \alpha; (n+1)^{-1}\left(\sum_{i=1}^{n} \delta_{V_i^y} + \delta_1\right)\right)\right\},$$

the probability of test-conditional miscoverage error can be written as

$$\mathbb{P}\left(Y_{n+1} \notin \widehat{C}_\alpha^{\text{GLCP}}(X_{n+1}) \mid X_{n+1} = x\right)$$
$$= \mathbb{P}\left(V_{n+1} > Q\left(1 - \alpha; (n+1)^{-1}\left(\sum_{i=1}^{n} \delta_{V_i} + \delta_1\right)\right) \mid X_{n+1} = x\right). \tag{12}$$

From equation (11), we let

$$\widehat{V}_q := Q\left(1 - \alpha; (n+1)^{-1}\left(\sum_{i=1}^{n} \delta_{V_i} + \delta_1\right)\right) = V_{(\lceil nq\rceil,n)},$$

where $q = \lceil (1-\alpha)(n+1)\rceil / n$. Notably, $\widehat{V}_q$ is the $q$-th quantile of empirical distribution $\widehat{F}_V = n^{-1}\sum_{i=1}^n \delta_{V_i}$. DKW inequality gives that for any $\delta \in (0,1)$:

$$\mathbb{P}\left(\sup_x |\widehat{F}_V(x) - F_V(x)| > \epsilon\right) \leq \delta,$$

where $\epsilon = \{(2n)^{-1}\ln(2/\delta)\}^{-1/2}$. The definition of quantile gives

$$q \leq \widehat{F}_V(\widehat{V}_q) < q + \frac{1}{n}.$$

Denote $A_\epsilon = \{\sup_x |\widehat{F}_V(x) - F_V(x)| \leq \epsilon\}$. On $A_\epsilon$ we have

$$q - \epsilon \leq F_V(\widehat{V}_q) < \epsilon + q + \frac{1}{n},$$
$$F_V^{-1}(q - \epsilon) \leq \widehat{V}_q < F_V^{-1}(\epsilon + q + n^{-1}).$$

Define $U_{n+1} = F(s(X_{n+1}, Y_{n+1}) \mid X_{n+1})$ with $X_{n+1} = x$, such that $U_{n+1} \sim U[0,1]$. Therefore, the total variation between the distribution of $V_{n+1}$ and $U_{n+1}$ is $\delta_1(x; \widehat{F})$. We have

$$
\begin{aligned}
(12) &\leq \delta_1(x; \widehat{F}) + \mathbb{P}\Big(U_{n+1} > Q\Big(1 - \alpha; (n+1)^{-1}\Big(\sum_{i=1}^{n} \delta_{V_i} + \delta_1\Big)\Big) \mid X_{n+1} = x\Big) \\
&\leq \delta_1(x; \widehat{F}) + \mathbb{P}(A_\epsilon^C) + \mathbb{P}\Big(A_\epsilon, U_{n+1} > \widehat{V}_q \mid X_{n+1} = x\Big) \\
&\leq \delta_1(x; \widehat{F}) + \delta + \mathbb{P}\Big(U_{n+1} > F_V^{-1}(q - \epsilon) \mid X_{n+1} = x\Big) \\
&= \delta_1(x; \widehat{F}) + \delta + 1 - F_V^{-1}(q - \epsilon).
\end{aligned}
$$

On the other hand, we can similarly derive that

$$
\begin{aligned}
(12) &\geq \mathbb{P}\Big(U_{n+1} > Q\Big(1 - \alpha; (n+1)^{-1}\Big(\sum_{i=1}^{n} \delta_{V_i} + \delta_1\Big)\Big) \mid X_{n+1} = x\Big) - \delta_1(x; \widehat{F}) \\
&\geq \mathbb{P}\Big(A_\epsilon, U_{n+1} > \widehat{V}_q \mid X_{n+1} = x\Big) - \delta_1(x; \widehat{F}) \\
&\geq \mathbb{P}\Big(A_\epsilon, U_{n+1} > F_V^{-1}(\epsilon + q + n^{-1}) \mid X_{n+1} = x\Big) - \delta_1(x; \widehat{F}) \\
&\geq \mathbb{P}\Big(U_{n+1} > F_V^{-1}(\epsilon + q + n^{-1}) \mid X_{n+1} = x\Big) - \mathbb{P}(A_\epsilon^C) - \delta_1(x; \widehat{F}) \\
&\geq 1 - F_V^{-1}(\epsilon + q + n^{-1}) - \delta_1(x; \widehat{F}) - \delta.
\end{aligned}
$$

Therefore,

$$
\begin{aligned}
&\big|\mathbb{P}\big(Y_{n+1} \notin \widehat{C}_\alpha^{\mathrm{GLCP}}(X_{n+1}) \mid X_{n+1} = x\big) - \alpha\big| \\
&\leq \max\big\{|1 - \alpha - F_V^{-1}(q - \epsilon) + \delta|, |1 - \alpha - F_V^{-1}(q + \epsilon + n^{-1}) - \delta|\big\} + \delta_1(x; \widehat{F}).
\end{aligned}
$$

### A.2.3 Proof of Theorem 2.8

**Part 1.**

For any $(x, s)$, based on the first inequality of Assumption 2, we have

$$
f_{\mathrm{mix}}(s \mid x) = \frac{\sum_{k=1}^{K} \pi_k f_k(x, s)}{\sum_{k=1}^{K} \pi_k f_k(x)} = \frac{\sum_{k=1}^{K} \pi_k f_k(x) f_k(s \mid x)}{\sum_{k=1}^{K} \pi_k f_k(x)} \geq L_K^{-1} \sum_{k=1}^{K} f_k(s \mid x).
$$

Moreover, based on the second property of Assumption 2, there exists $k \in [K]$ such that $f_k(s \mid x) \geq f(s \mid x)/L_0$. Therefore, considering Assumption 1, we have

$$
f_{\mathrm{mix}}(s \mid x) \geq (L_0 L_K)^{-1} f(s \mid x) \geq \{L_f(x) L_0 L_K\}^{-1}.
$$

Based on Assumption 3, we denote

$$
s_x(\widehat{r}) = \operatorname*{arg\,max}_{s \in [0, M_1(x)]} \widehat{r}(x, s).
$$

Based on the Lipschitz property we can derive that

$$
\begin{aligned}
\int_0^\infty \widehat{r}(x, t)dt &\geq \int_{\{s_x(\widehat{r}) - \widehat{r}(x, s_x(\widehat{r}))/L(\widehat{r}, x)\} \vee 0}^{s_x(\widehat{r}) + \widehat{r}(x, s_x(\widehat{r}))/L(\widehat{r}, x)} \widehat{r}(x, t)dt \\
&\geq \int_0^{\widehat{r}(x, s_x(\widehat{r}))/L(\widehat{r}, x)} L(\widehat{r}, x)t \, dt = \frac{\{\widehat{r}(x, s_x(\widehat{r}))\}^2}{2L(\widehat{r}, x)}.
\end{aligned}
$$

Therefore

$$
\widehat{r}(s \mid x) = \frac{\widehat{r}(x, s)}{\int_0^\infty f_{\mathrm{mix}}(t \mid x)\widehat{r}(x, t)dt} \leq \frac{2L_f(x) L_0 L_K L(\widehat{r}, x)}{\widehat{r}(x, s_x(\widehat{r}))},
$$

which does not rely on $s$ and $\delta_2(x; \widehat{F}_{\mathrm{mix}}) < \infty$.

**Part 2.**

In the following proof we assume the conditional distribution estimators $\widehat{F}(s \mid x)$ and $\widehat{F}_{\mathrm{mix}}(s \mid x)$ are continuous with respect to $s$ and differentiable. The derivative with respect to $s$ is denoted as conditional density estimators $\widehat{f}(s \mid x)$ and $\widehat{f}_{\mathrm{mix}}(s \mid x)$. We rewrite the aggregated conditional distribution estimator as follows:

$$\widehat{F}_{\mathrm{agg}}(s \mid x) = \{1 + \widehat{r}(x)\}^{-1} \int_0^s \widehat{f}(t \mid x) + \widehat{f}_{\mathrm{mix}}(t \mid x)\widehat{r}(t \mid x)\widehat{r}(x)dt \,,$$

which is derived by

$$F(s \mid x) = \int_0^s f(t \mid x)dt = \int_0^s f_{\mathrm{mix}}(t \mid x)r(t \mid x)dt \,,$$

and weight two estimators with 1 and $\widehat{r}(x)$.

Notably, the total variation can be written as the integration of the abstract difference of densities:

$$\delta_1(x; \widehat{F}) = \frac{1}{2} \int_0^\infty \big|\widehat{f}(s \mid x) - f(s \mid x)\big|ds \,,$$

$$\delta(x; \widehat{F}_{\mathrm{mix}}, F_{\mathrm{mix}}) = \frac{1}{2} \int_0^\infty \big|\widehat{f}_{\mathrm{mix}}(s \mid x) - f_{\mathrm{mix}}(s \mid x)\big|ds \,.$$

The derivative of $\widehat{F}_{\mathrm{agg}}(s \mid x)$ with respect to $s$ is

$$\widehat{f}_{\mathrm{agg}}(s \mid x) = \{1 + \widehat{r}(x)\}^{-1}\big\{\widehat{f}(s \mid x) + \widehat{f}_{\mathrm{mix}}(s \mid x)\widehat{r}(s \mid x)\widehat{r}(x)\big\} \,.$$

Thus the $\delta_1(x; \widehat{F}_{\mathrm{agg}})$ can be calculated as

$$
\begin{aligned}
\delta_1(x; \widehat{F}_{\mathrm{agg}}) &= \int_0^\infty \big|\widehat{f}_{\mathrm{agg}}(s \mid x) - f(s \mid x)\big|ds \\
&\leq \{1 + \widehat{r}(x)\}^{-1}\Big\{ \int_0^\infty \big|\widehat{f}(s \mid x) - f(s \mid x)\big|ds + \\
&\qquad\qquad \widehat{r}(x) \int_0^\infty \big|\widehat{f}_{\mathrm{mix}}(s \mid x)\widehat{r}(s \mid x) - f(s \mid x)\big|ds\Big\} \\
&= \{1 + \widehat{r}(x)\}^{-1}\Big\{2\delta_1(x; \widehat{F}) + \widehat{r}(x) \int_0^\infty \big|\widehat{f}_{\mathrm{mix}}(s \mid x)\widehat{r}(s \mid x) - f(s \mid x)\big|ds\Big\} \\
&= \{1 + \widehat{r}(x)\}^{-1}\Big\{2\delta_1(x; \widehat{F}) + \\
&\qquad\qquad \int_0^\infty \big|\widehat{f}_{\mathrm{mix}}(s \mid x)\widehat{r}(x, s) - f_{\mathrm{mix}}(s \mid x)r(x, s)\frac{\widehat{r}(x)}{r(x)}\big|ds\Big\} \\
&\leq \{1 + \widehat{r}(x)\}^{-1}\Big\{2\delta_1(x; \widehat{F}) + \int_0^\infty \big|\widehat{f}_{\mathrm{mix}}(s \mid x)\widehat{r}(x, s) - f_{\mathrm{mix}}(s \mid x)r(x, s)\big|ds + \\
&\qquad\qquad \int_0^\infty \big|f_{\mathrm{mix}}(s \mid x)r(x, s) - f_{\mathrm{mix}}(s \mid x)r(x, s)\frac{\widehat{r}(x)}{r(x)}\big|ds\Big\} \\
&\leq \{1 + \widehat{r}(x)\}^{-1}\Big[2\delta_1(x; \widehat{F}) + \int_0^\infty f(s \mid x)r(x)\big|1 - \widehat{r}(x)/r(x)\big|ds + \\
&\qquad\qquad \int_0^\infty \big|f_{\mathrm{mix}}(s \mid x)\{\widehat{r}(x, s) - r(x, s)\}\big|ds + \\
&\qquad\qquad \int_0^\infty \big|\widehat{r}(x, s)\{\widehat{f}_{\mathrm{mix}}(s \mid x) - f_{\mathrm{mix}}(s \mid x)\}\big|ds\Big] \\
&\leq \{1 + \widehat{r}(x)\}^{-1}\Big(2\delta_1(x; \widehat{F}) + \big|r(x) - \widehat{r}(x)\big| + \widetilde{\delta}(x; \widehat{r}) + \\
&\qquad\qquad \Big[\int_0^\infty \widehat{r}^2(x, s)ds \int_0^\infty \{\widehat{f}_{\mathrm{mix}}(s \mid x) - f_{\mathrm{mix}}(s \mid x)\}^2 ds\Big]^{-1/2}\Big) \\
&= \{1 + \widehat{r}(x)\}^{-1}\Big[2\delta_1(x; \widehat{F}) + \widetilde{\delta}(x; \widehat{r}) + \big|\widehat{r}(x) - r(x)\big| + \delta_2(x; \widehat{F}_{\mathrm{mix}})L_2(x; \widehat{r})\Big] \,.
\end{aligned}
$$

### A.3 One-Shot Agent Similarity Matching

In real-world federated learning scenarios where the number of source agents can scale to thousands (e.g., smart device ecosystems with thousands of user-level agents), two critical challenges arise: (1) the computational infeasibility of utilizing all agents, and (2) the potential presence of distributional outliers that degrade model performance. Our framework addresses both issues through efficient agent selection—first reducing computational costs by prioritizing top-$K'$ agents via lightweight distributional alignment metrics. We test our similarity metric in Section A.4.4

#### A.3.1 Federated Data Similarity: A Survey

In federated learning for classification tasks, a common approach measures agent similarity by computing class label distributions (or marginal response distributions in regression settings) as feature vectors, then comparing these vectors across agents [7]. However, this method oversimplifies distributional alignment by solely considering label/response statistics, failing to account for the joint distributional relationships between inputs $X$ and outputs $Y$. Such marginal matching proves insufficient when covariate shifts or feature-label dependency mismatches exist across agents.

Briggs et al. [3], Fraboni et al. [10] proposed gradient-similarity-aware clustering. Here, clients are grouped by the similarity of their model weight updates compared to the global joint model [3], or representative gradients [10], defined as the difference between their locally updated model parameters and the global model. These gradients implicitly capture the direction and magnitude of client-specific updates, reflecting underlying data distributions. A hierarchical clustering algorithm partitions clients into clusters such that those within a cluster share similar gradient profiles, ensuring that each cluster represents a distinct data pattern.

Existing gradient-based methods for measuring similarity among federated agents require participation in global task training, which imposes substantial computational overhead. However, in our framework, we propose a lightweight alternative that directly screens source agents during preliminary analysis to accelerate similarity quantification. This paradigm shift enables rapid identification of inter-agent correlations without full model deployment, permitting selectively utilizing a subset of source agents with higher similarity for model training. We subsequently introduce an efficient Kmeans-driven similarity metric that operates on parameter distributions rather than gradient interactions, achieving comparable discriminative power while significantly reducing time complexity.

#### A.3.2 K-means-Driven Similarity Metric

A classical approach for measuring distributional similarity [4, 2] between two datasets $D_1$ and $D_2$ involves binning the data into $M$ intervals. This produces histogram vectors $\Pi_1 = (\pi_{11}, \ldots, \pi_{1M})$ and $\Pi_2 = (\pi_{21}, \ldots, \pi_{2M})$, where $\pi_{im}$ represents the proportion of data points from $D_i$ falling into the $m$-th bin. Standard bin-to-bin similarity measures can then be applied to quantify distribution alignment based on these frequency vectors. We list several classical measurements here:

1. **Total Variation:** This index measures the $L_1$ distance of two input bins:

$$d_{\mathrm{TV}}(\Pi_1, \Pi_2) = \frac{1}{2} \sum_{i=1}^{M} |\pi_{1i} - \pi_{2i}| .$$

2. **Histogram Intersection:** This index is expressed by a $\min$ function to get the overlap of two input bins:

$$d_{\mathrm{Inter}}(\Pi_1, \Pi_2) = 1 - \sum_{i=1}^{M} \min(\pi_{1i}, \pi_{2i}) .$$

3. **Histogram Correlation:** This index measures the correlation of two input bins:

$$d_{\mathrm{Corr}}(\Pi_1, \Pi_2) = \frac{\sum_{i=1}^{M}(\pi_{1i} - \overline{\pi}_1)(\pi_{2i} - \overline{\pi}_2)}{\sqrt{\sum_{i=1}^{M}(\pi_{1i} - \overline{\pi}_1)^2 \sum_{i=1}^{M}(\pi_{2i} - \overline{\pi}_2)^2}} .$$

Existing approaches for dataset similarity assessment in federated learning often rely on raw data interactions, posing privacy risks and communication bottlenecks. We address this by developing a privacy-preserving binning strategy: instead of exchanging original data samples, all agents collaboratively construct a global codebook through K-means clustering on aggregated data distributions, partitioning the feature space into $M$ non-overlapping subspaces where each point belongs to its nearest cluster center. By converting local datasets into cluster-indexed histograms through simple counting operations, agents can efficiently compute distribution similarities using histogram distances.

Following the established federated clustering paradigm [11], each source agent $k \in \{1, ..., K\}$ initially identifies $M_0$ local cluster centers $\{C_k^{(1)}, ..., C_k^{(M_0)}\}$ through localized K-means clustering on its dataset $\mathcal{D}_k$, while recording corresponding sample counts $\{N_k^{(1)}, ..., N_k^{(M_0)}\}$. These locally optimized cluster centers and their associated cardinalities are securely transmitted to the central server, which aggregates all $K \times M_0$ centers and performs weighted K-means clustering – using the transmitted sample counts as importance weights – to derive $M$ global cluster centroids. Upon receiving these consensus cluster centroids, each source agent constructs a distribution histogram by mapping its local samples to the nearest cluster centroids. The histogram of source agent $k$ is denoted by $\Pi_k = (\pi_{k1}, \ldots, \pi_{kM_0})$. Therefore we measure the distance of $\mathcal{D}_i$ and $\mathcal{D}_j$ by

$$d_{\mathrm{Km}}(\mathcal{D}_i, \mathcal{D}_j) = d_{\mathrm{TV}}(\Pi_i, \Pi_j) .$$

## A.4 Additional Experiments: Synthetic Data

### A.4.1 Experiments on Different Extent of Source Agent Data Heterogeneity

Our second experiment quantifies how varying degrees of source agent heterogeneity ($\gamma_k$ distribution shifts) affect target task performance. We design two regimes: **S1.3 & S2.3:** $\gamma_k \sim U[1 - c_1, 1]$ and **S1.4 & S2.4:** $\gamma_k \sim U[1, 1 + c_2]$, where $c_1 \in [0, 1]$ and $c_2 \in [0, 5]$ control heterogeneity intensity (larger values indicate stronger distributional mismatch). Density ratio estimation is fixed to the calibrated classifier approach to isolate heterogeneity effects. This setup systematically probes how under-provisioned ($c_1$-driven) versus over-provisioned ($c_2$-driven) source agent distributions influence federated conformal prediction robustness.

Figure 4 demonstrates that PFCP exhibits notable robustness to distributional heterogeneity, achieving consistent performance improvements compared to baseline methods across a broad spectrum of $\gamma_k$ variation magnitudes. This empirically verifies PFCP's capability to leverage useful source agents while automatically discounting those with severely mismatched distributions. Furthermore, the performance degradation remains gradual even under extreme heterogeneity, highlighting practical reliability in real-world federated scenarios where precise prior knowledge of agent distributions is often unavailable.

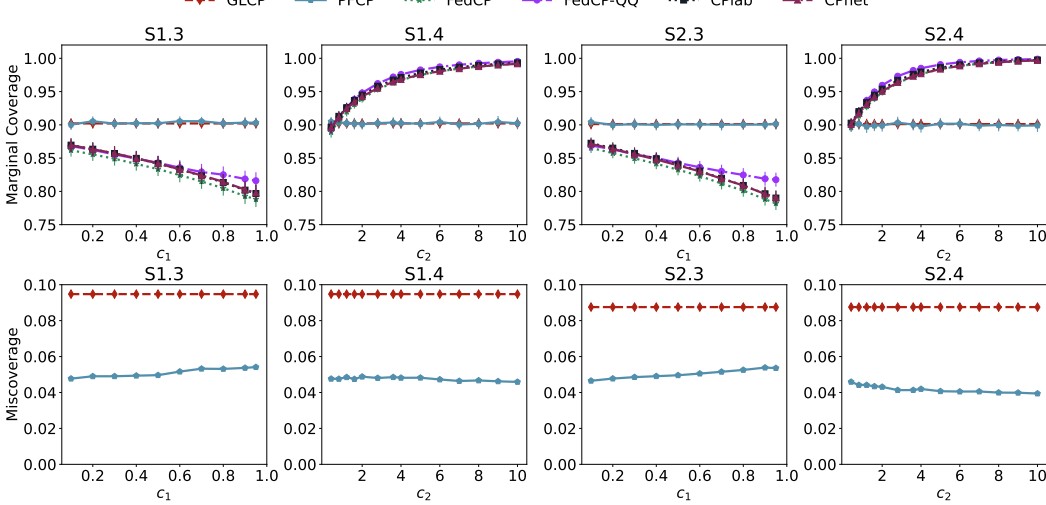

Figure 4: Results of synthetic data with varying source agent distribution.

### A.4.2 Experiments on Different Source Agents Quantities

Central to PFCP is leveraging auxiliary data from source agents to enhance data-scarce target tasks. Follow the setting of **S1** and **S2** and denote the setting with $\gamma_k \sim U[0.6, 1]$ as **S1.5** and **S2.5**, we systematically investigate how performance scales with increasing numbers of source agents ($K \in \{2, 4, \ldots, 20\}$). Each agent, including the target, maintains identical data volume ($3n$ samples: for the source agents $n$ for point estimation, $2n$ for conditional distribution calibration). This controlled experiment isolates the effect of collaborative scale, revealing an empirical scaling law where PFCP's prediction interval width decreases sublinearly with $K$, achieving improvements in sample efficiency over single-agent baselines.

Figure 5 demonstrates that even minimal collaboration (2 source agents) delivers substantial improvements–reducing conditional miscoverage by 19.7% (S1.5) and 25.6% (S2.5), and prediction interval widths by 10.5% (S1.5) and 3.9% (S2.5) compared to isolated learning. The enhancements scale sublinearly with agent count $K$. This quantifies PFCP's data-efficiency in leveraging sparse yet distributed knowledge sources.

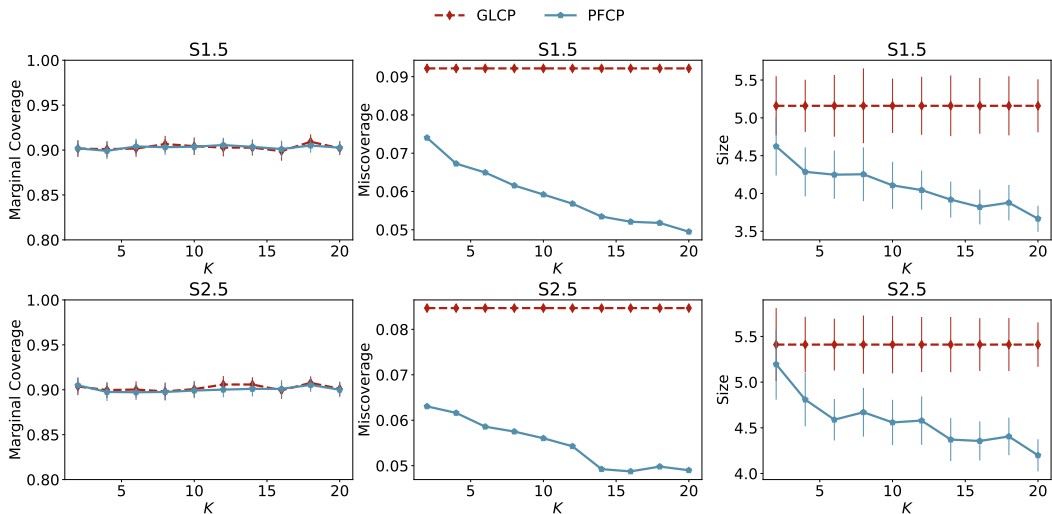

Figure 5: Results of synthetic data with varying source agent quantities.

### A.4.3 Experiments on Different Required Coverage Levels

To rigorously assess PFCP's robustness beyond the conventional $\alpha = 0.1$ setting, we systematically evaluate its performance across $\alpha \in [0.05, 0.95]$. We still follow the setting of **S1** and **S2** and denote the varying $\alpha$ setting as **S1.6** and **S2.6**, where we set $\gamma_k \sim U[0.6, 1]$.

Figure 6 reveals that GLCP exhibits slight over-coverage tendencies at extreme $\alpha$ values ($\alpha \geq 0.9$), due to instability in prediction set construction under sparse calibration scenarios. In contrast, PFCP maintains coverage rates tightly clustered around the target $1 - \alpha$ level across the entire spectrum. Also Figure 6 reveals that PFCP maintains superior reliability over GLCP throughout this parametric spectrum: while both methods achieve nominal marginal coverage, PFCP reduces conditional miscoverage across extreme quantiles ($\alpha < 0.3$ or $\alpha > 0.8$) through adaptive density ratio reweighting. Simultaneously, the prediction interval size analysis demonstrates that while both methods show monotonically increasing interval lengths with higher $1 - \alpha$ requirements, PFCP achieves systematically tighter intervals.

This systematic evaluation demonstrates PFCP's unique capability to preserve both marginal validity and conditional calibration across diverse coverage requirements while achieving dual improvements: exact coverage maintenance and statistically efficient interval growth. By adaptively balancing reliability and precision, PFCP enhances statistical efficiency without compromising robustness–a critical advantage for real-world applications demanding flexible yet rigorous error rate control.

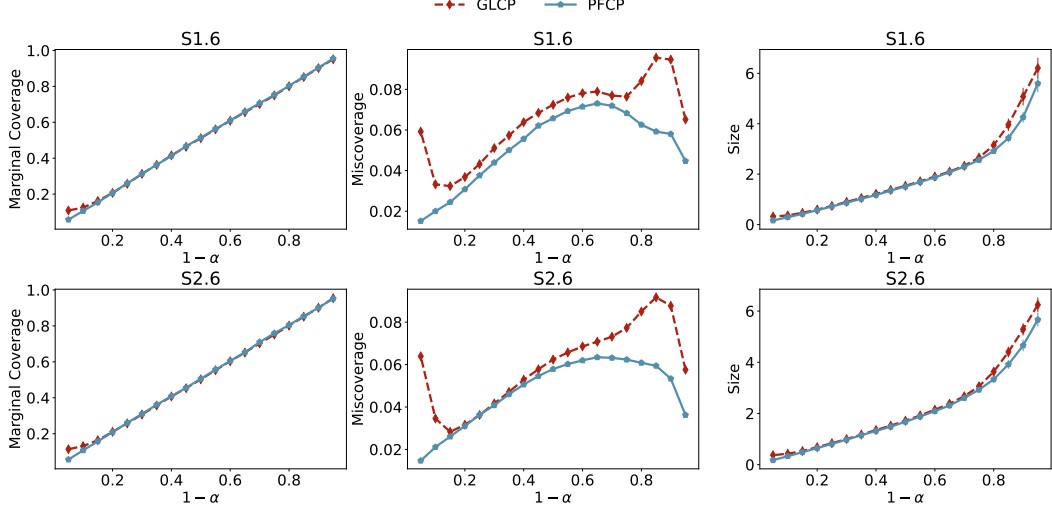

Figure 6: Results of synthetic data with varying required coverage.

### A.4.4 Efficiency of K-means-Driven Similarity Metric: A Toy Example

In this toy example, we consider 40 source agents with $p_{\text{cor}}$-controlled corruption levels, where each agent's distribution similarity parameter $\gamma_k$ follows the mixture model $\gamma_k \sim p_{\text{cor}}U[0, 0.2] + (1 - p_{\text{cor}})U[0.6, 1]$. Following **S1** and **S2**, two experimental scenarios (**S1.7** & **S2.7**) are designed to simulate varying proportions of corrupted agents exhibiting distributional divergence from the target domain ($\gamma = 1$). The ranking efficacy of similarity metric $d_{\text{Km}}$ is quantified by verifying whether the sorted agent sequence maintains decreasing $\gamma_k$ values – monotonic ordering indicates that $d$ successfully prioritizes agents with higher distributional alignment to the target.

For each repeat, we denote the order statistics of $\gamma_1, \ldots, \gamma_K$ by $\gamma_{(1)} \geq \ldots \geq \gamma_{(K)}$. The corresponding similarity parameter of the sorted agent sequence is denoted by $\widetilde{\gamma}_1, \ldots, \widetilde{\gamma}_K$. First we compare $\mathbb{E}\gamma_{(1)}, \ldots, \mathbb{E}\gamma_{(K)}$ and $\mathbb{E}\widetilde{\gamma}_1, \ldots, \mathbb{E}\widetilde{\gamma}_K$ in Figure 7. The experimental results demonstrate that our similarity metric achieves effective distributional discrimination, as evidenced by the monotonic ranking of source agents – those prioritized by the method exhibit significantly closer alignment to the target distribution ($\gamma = 1$). Notably, the curve reveals a steep decline phase where our selective screening coincides perfectly with the oracle selection, indicating that prior knowledge of the corruption ratio $p_{\text{cor}}$ enables near-perfect identification of distributionally divergent agents through this approach.

When utilizing filtered source agents as an aggregated composite distribution, next we focus on evaluating the collective alignment through average similarity parameters. Specifically, upon selecting the top $K'$ agents ranked by $d_{\text{Km}}$, we quantify their ensemble relevance by computing $\bar{\gamma}_{K'} = \sum_{k=1}^{K'} \widetilde{\gamma}_k / K'$. We compare $\bar{\gamma}_{K'}$ with optimal $\widehat{\gamma}_{K'} = \sum_{k=1}^{K'} \gamma_{(k)} / K'$ in Figure 8.

### A.4.5 Efficiency of K-means-Driven Similarity Metric

We test the performance of PFCP with similarity agent matching using setting **S1.7** and **S2.7**. We consider 40 source agents and select top $n_{\text{sel}}$ to serve as the auxiliary agents. We test $p_{\text{cor}} = 0.1, 0.3, 0.5, 0.7, 0.9$ and $n_{\text{sel}} = 10, 20, 30$ respectively.

Figure 9 demonstrates the performance across four methods under fixed selected agent numbers $n_{\text{sel}}$, where the marginal coverage of FedCP and FedCP-QQ progressively degrade as the corrupted agent ratio $p_{\text{cor}}$ increases. These methods exhibit increasing susceptibility to distributionally divergent agents at higher corruption levels.

Figure 10 illustrates the conditional miscoverage dynamics of PFCP under varying corrupted ratios $p$, demonstrating that our similarity-aware matching mechanism effectively desensitizes the algorithm to distributional outliers. The density ratio estimation intrinsically mitigates adverse

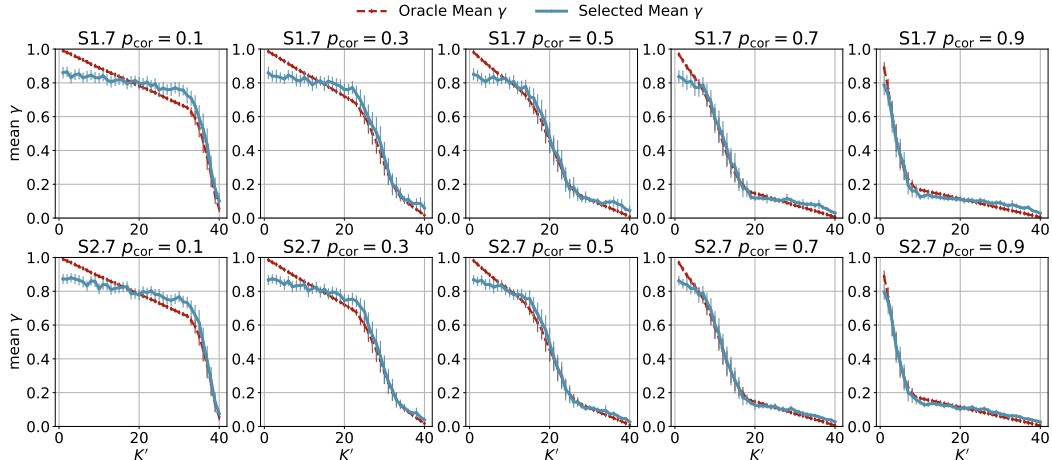

Figure 7: Similarity of the $k$-th selected agent.

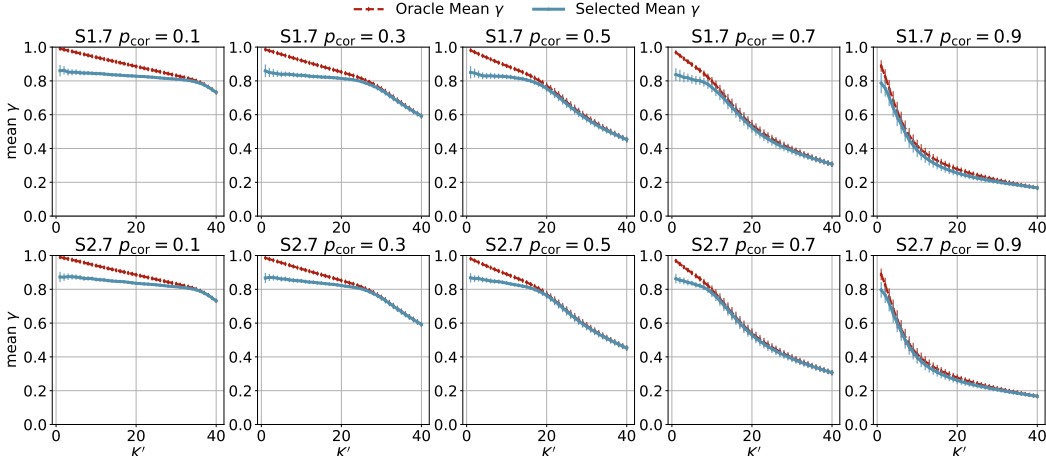

Figure 8: Average similarity of the top $k$ selected agent.

impacts from corrupted agents by reweighting samples according to their distributional congruence with the target domain. Complementing these findings, Figure 11 confirms PFCP's robustness in high-corruption regimes – even under severe data erosion, incremental improvements in predictive reliability emerge through expanded dataset aggregation, though such gains remain statistically marginal due to fundamental limitations in information recovery from corrupted sources.

### A.5   Additional Results of Real Data

This section presents the agent partitioning strategy and supplementary visual analyses omitted from the main text, including conditional coverage uniformity across feature subspaces for the BIO, BIKE, CRIME, and STAR datasets, and the scaling laws of transfer efficiency with respect to the number of source agents on BIO, BIKE, CRIME, STAR, and CONCRETE datasets, quantifying the diminishing returns of agent aggregation under domain heterogeneity.

**Agent Partitioning Strategy**: For the BIO dataset (feature dimension $d = 9$, response: Size of the residue) and the CONCRETE dataset ($d = 8$, response: Concrete compressive strength), the features themselves do not exhibit clear distinctions for agent partitioning. However, domain-specific scenarios may lead to systematic variations—for instance, certain protein studies may involve residues with atypically large sizes, while some concrete manufacturers may produce materials with lower compressive strength due to varying equipment or processes. To capture such distributional

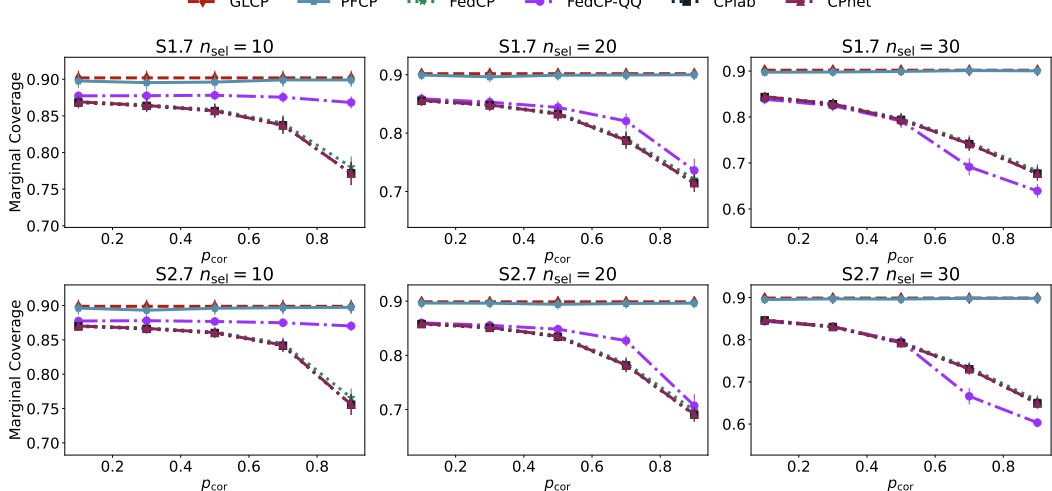

Figure 9: Marginal coverage with varying $p_{\text{cor}}$.

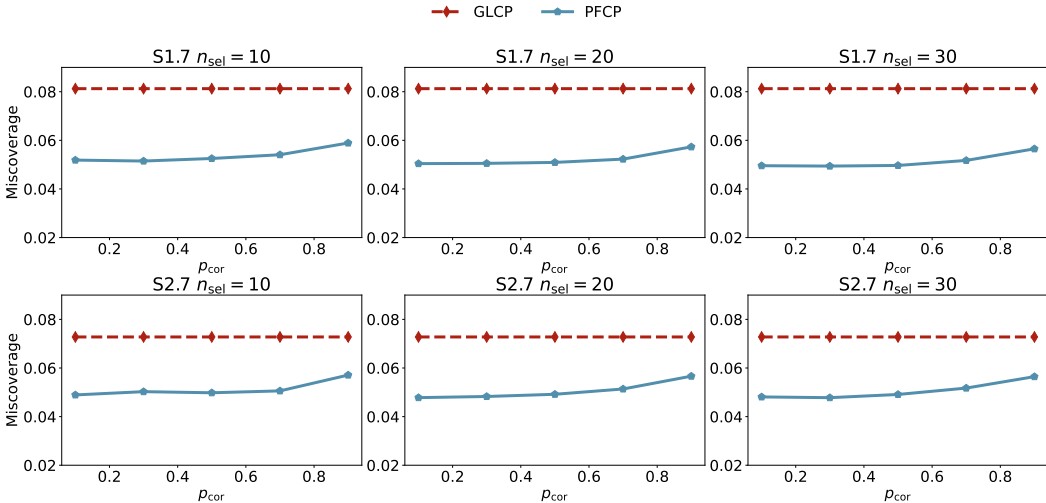

Figure 10: Conditional miscoverage with varying $p_{\text{cor}}$.

differences, we adopt a response-based partitioning scheme. Given a dataset $\{(x_1, y_1), \ldots, (x_n, y_n)\}$, we first compute the $\alpha_0$-quantile $y_0$ of the response variable. Each sample $(x_i, y_i)$ is then assigned a weight proportional to $K(|y_i - y_0|/\sigma_0)$, where $K(\cdot)$ is the Gaussian kernel function and $\sigma_0$ scales the deviation. Finally, we perform probability-proportional sampling without replacement to construct the agent-specific dataset. To induce meaningful distribution shifts across agents, we set the quantile parameter $\alpha_0$ to $0.9$ for the Protein dataset and $0.3$ for the Concrete dataset. The scale parameter $\sigma_0$ is defined as the difference between the $(\alpha_0 + 0.1)$ and $(\alpha_0 - 0.1)$ quantiles of the response variable, ensuring adaptive bandwidth for kernel weighting.

For the BIKE dataset (feature dimension $d = 9$, response: user count per record), we partition the data by month since it is recorded chronologically, using the last four months as the target agent and each remaining month as a separate agent. For the CRIME dataset ($d = 20$, response: violent crimes per 100K population per district), where existing research primarily focuses on densely populated metropolitan areas leaving less-populated regions understudied, we designate the district with the smallest population as the target agent and randomly partition the others. In the STAR dataset ($d = 20$, response: ACT score), given that many studies examine the impact of early education on

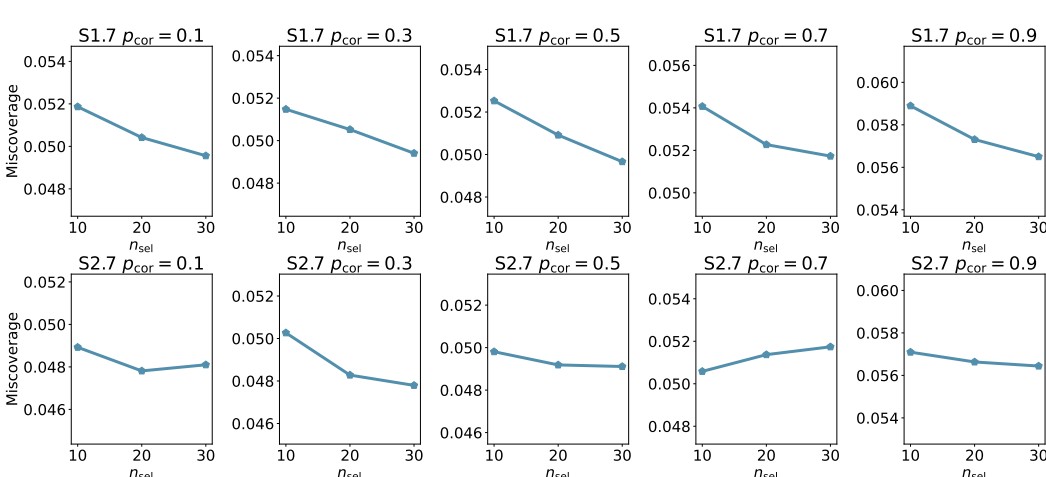

Figure 11: Conditional miscoverage with varying $n_{\text{sel}}$.

future achievement - particularly the negative effects of resource deprivation - we define students from rural schools in their early age as the target agent and randomly partition the remaining students.

For the classification dataset DERMA, the input image format for each sample is $28 \times 28 \times 3$. There are a total of 7 classes, and the class distribution is highly imbalanced, with some classes accounting for a very small proportion (e.g., some specific symptoms are rare). To simulate the prediction of disease susceptibility for a particular high-risk group, where such a group contains a relatively higher proportion of rare classes, we sample from all classes as uniformly as possible to form the target data pool, while the remaining samples constitute the auxiliary data pool. When partitioning the auxiliary data, it is divided evenly according to the number of agents.

**Conditional Coverage Calculation**: Computing conditional coverage for prediction sets requires knowing the exact conditional distribution of $Y$ given $X$, which is unavailable in real-world experiments. For each test sample $(X, Y)$ and prediction set $\widehat{C}_\alpha(X)$, we evaluate coverage via the indicator $\mathbb{1}(Y \in \widehat{C}_\alpha(X))$. We evaluate our method using relaxed conditional coverage. We partition the feature space into $N_x$ disjoint subspaces, ensuring each test sample belongs to exactly one subspace. Over 100 trials, we compute the average coverage rate within each subspace as the conditional coverage metric. Let $p_1, \ldots, p_{N_x}$ denote the conditional coverage rates and $w_1, \ldots, w_{N_x}$ the proportions of samples in each subspace. The conditional miscoverage is then defined as $\sum_{i=1}^{N_x} w_i |p_i - (1 - \alpha)|$. The partitioning strategy in this work first applies K-means clustering to the target agent's data to identify $N_x$ centroids, then divides the spatial domain into $N_x$ disjoint sub-regions based on these centroids, ensuring spatially coherent and interpretable partitions while maintaining alignment with the underlying data distribution.

For classification datasets, the test-conditional coverage is directly partitioned based on the class labels of the samples. Assume there exist $N_y$ classes for the dataset, and let $p_1, \ldots, p_{N_y}$ denote the conditional coverage rates and $w_1, \ldots, w_{N_y}$ the proportions of samples in each class. The gap between the coverage and $1 - \alpha$ is computed separately for each class, and then weighted by the proportion of each class to obtain the conditional miscoverage $\sum_{i=1}^{N_y} w_i |p_i - (1 - \alpha)|$.

## A.6 Limitations

The method is designed to enhance conformal prediction in a federated setting by leveraging auxiliary data from multiple source agents to support conformal prediction on the target agent. While effective under certain conditions, this strategy assumes that the data distributions of the source and target agents are relatively aligned. If the distributional shift between source and target is extremely significant, the effectiveness of the auxiliary data may degrade. The estimated $\widehat{r}$ can be approximately 0 on the support of source agent data and makes source agent data useless.

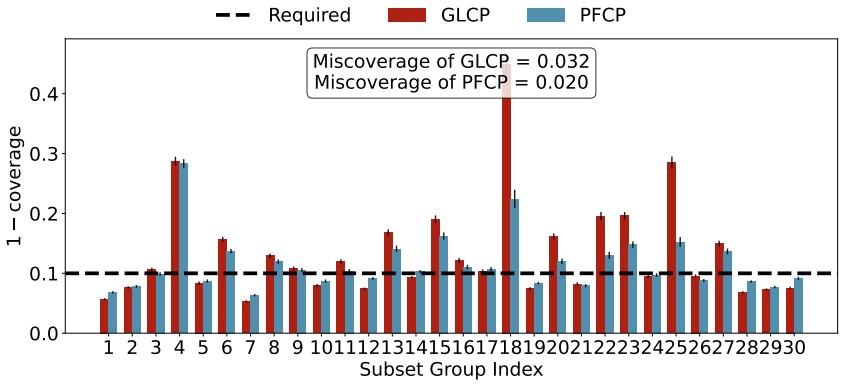

Figure 12: Conditional coverage over all subspace for BIO dataset.

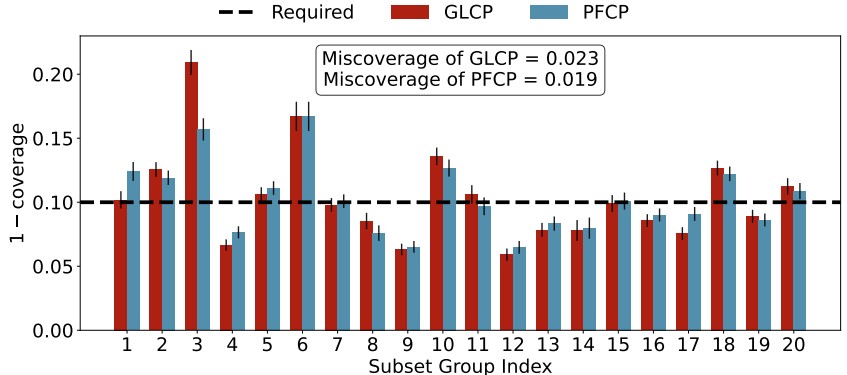

Figure 13: Conditional coverage over all subspace for BIKE dataset.

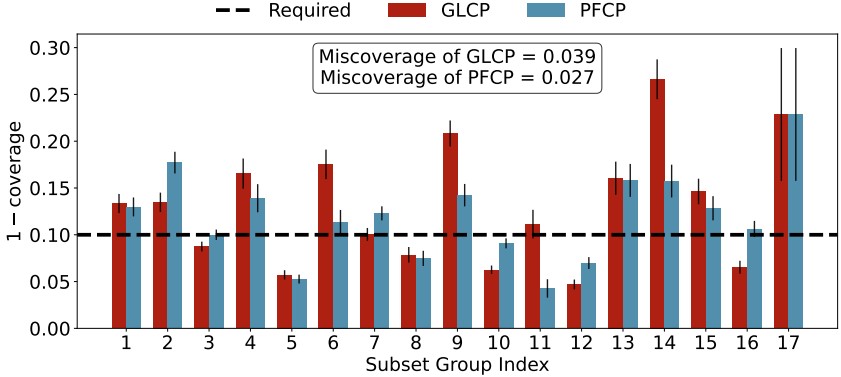

Figure 14: Conditional coverage over all subspace for CRIME dataset.

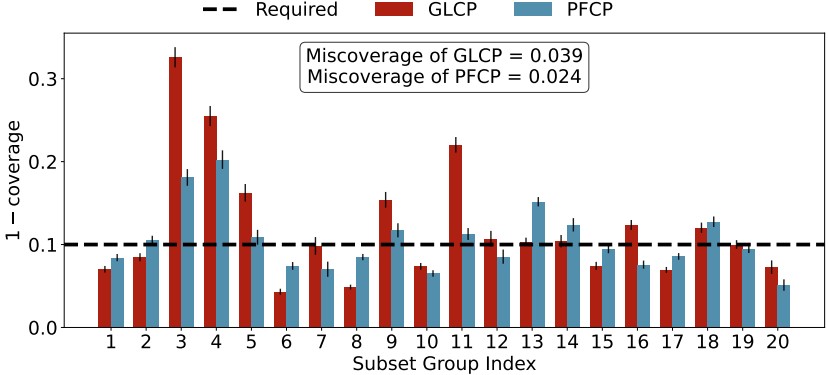

Figure 15: Conditional coverage over all subspace for STAR dataset.

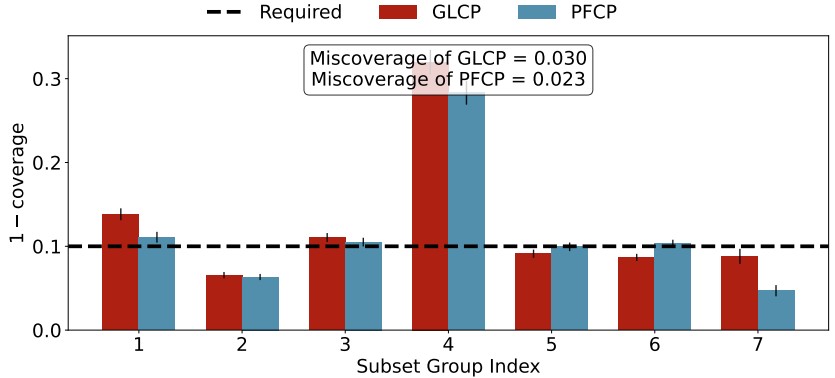

Figure 16: Conditional coverage over all subspace for DERMA dataset.

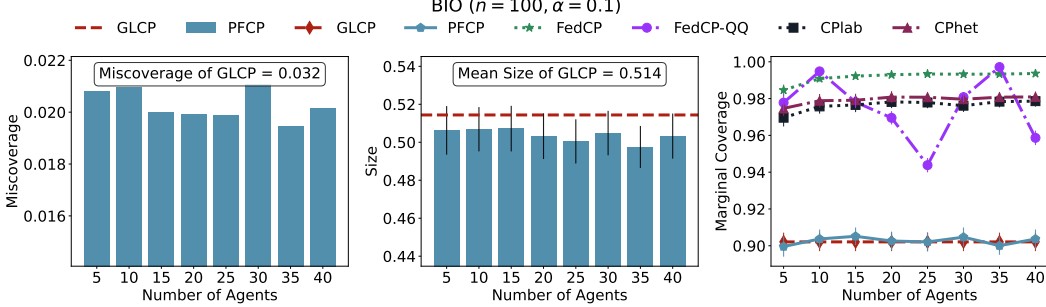

Figure 17: Impact of source agent quantity on target task performance for BIO dataset.

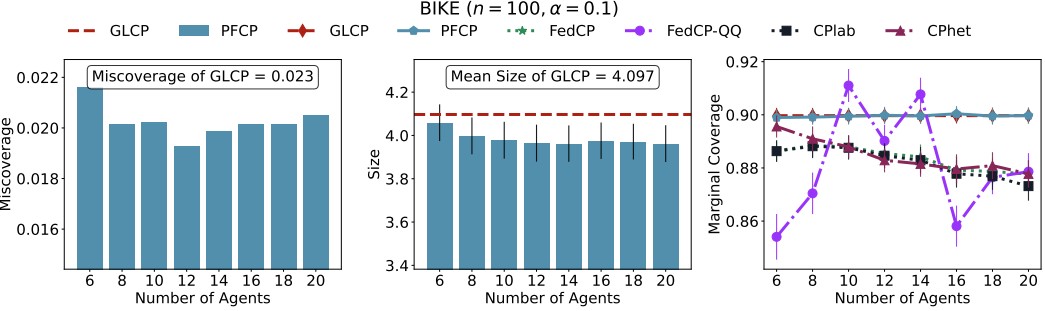

Figure 18: Impact of source agent quantity on target task performance for BIKE dataset.

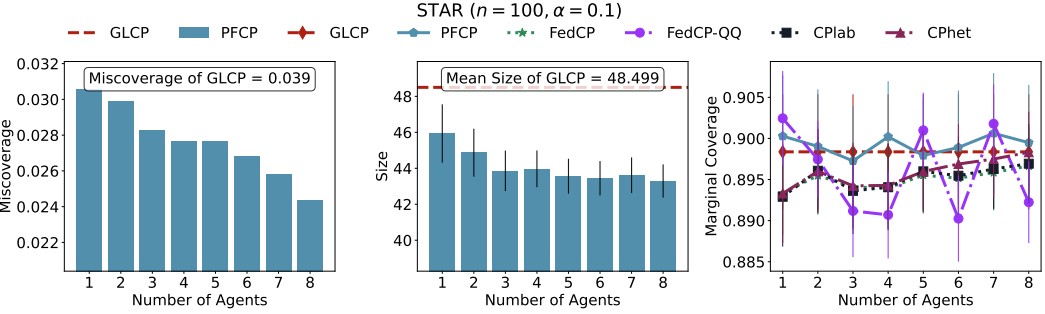

Figure 19: Impact of source agent quantity on target task performance for STAR dataset.

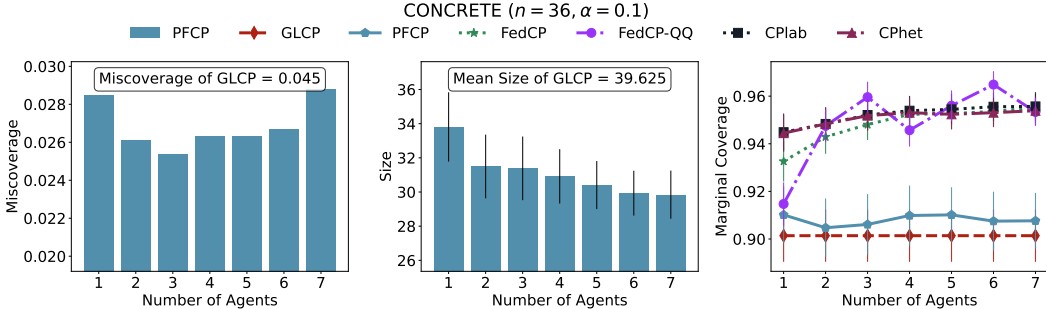

Figure 20: Impact of source agent quantity on target task performance for CONCRETE dataset.

Moreover, incorporating a large volume of auxiliary data from multiple sources introduces significant computational overhead. This includes communication costs in federated settings and increased computational requirements for estimating $\widehat{F}_{\text{agg}}$. Future work could address these issues by developing adaptive strategies that weigh source contributions based on distributional similarity or by designing more efficient protocols for federated uncertainty quantification.

## A.7 Societal Impact

The methods proposed have implications that can be interpreted as having both positive and negative societal consequences. On the positive side, the proposed conformal prediction methods enable more reliable uncertainty quantification in complex predictive tasks. This has the potential to significantly enhance decision-making in high-stakes domains such as healthcare, finance, and environmental modeling, where accurate assessments of predictive uncertainty are crucial. Furthermore, by improving the reliability and fairness of predictions, these methods may help foster greater trust in AI systems, especially when applied in sensitive settings involving diverse population groups.

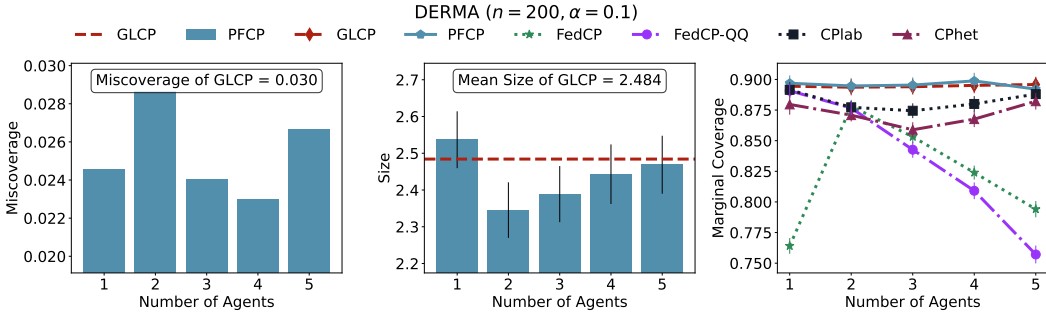

Figure 21: Impact of source agent quantity on target task performance for DERMA dataset.

On the other hand, there are potential negative impacts to consider. A key concern is that overreliance on conformal prediction techniques–particularly without sufficient domain expertise–may lead to overly simplistic or misinformed interpretations of the uncertainty information provided. This could result in suboptimal or even harmful decisions if the limitations of the method are not fully understood or communicated. As such, while the work contributes valuable tools for uncertainty quantification, its responsible deployment requires careful consideration of context, assumptions, and the broader social environment in which such tools are used.

