# OpenReview forum: "Personalized Federated Conformal Prediction with Localization"
_NeurIPS.cc/2025/Conference — NeurIPS 2025 poster_

### Official Review · Reviewer_2MYP · 2025-06-15

**Clarity:** 3
**Significance:** 3
**Originality:** 3
**Rating:** 4
**Confidence:** 4

**Summary:**

This paper considers the setup of federated learning, where the data comes from multiple agents, and the goal is to quantify the uncertainty of a test agent. The challenge is that the source and target agents have different distributions. This work proposes “personalized federated conformal prediction”, a method that constructs prediction sets that contain the responses of the test agent at a user-specified probability. This method builds on conformal prediction and localized conformal prediction. The theoretical analysis indicates that the proposed aggregation of conditional distribution estimators should attain better conditional coverage compared to the baseline Generalized Localized Conformal Prediction. Experiments on synthetic and real datasets reveal that the proposed PFCP performs better than prior work.

**Questions:**

## Questions
1.	In line 200, what does “valid” estimator mean?
2.	The informativeness of the bounds in Lemma 2.6 and Theorem 2.8 is not clear. Is it possible that the bounds might get trivial in practice?
3.	In line 249, $\tilde{\delta}$ is defined as an integral over $x$ or $s$?
4.	In line 266, over what the randomness is taken? Random data splits?
5.	How is the test-conditional miscoverage computed in the experiments?

## Minor comments
1.	The proof for Lemma 2.1 follows directly from the theory of conformal prediction, as in [1], but it is not stated in the paper. Since there is no proof for this Lemma, it should be at least noted that it follows from general results on conformal inference.
2.	The weight $\pi$ is defined in line 111, but it is only used in line 147. It should be defined close to its first use for clarity.

**Ethical Concerns:**

["NO or VERY MINOR ethics concerns only"]

**Final Justification:**

The authors conducted additional experiments that highlight the advantages of their proposed approach.

**Limitations:**

Yes.

**Paper Formatting Concerns:**

No.

**Quality:**

3

**Strengths And Weaknesses:**

## Strengths
1.	The proposed PFCP attains better performance compared to the baseline GLCP and prior work.
2.	This paper provides a theoretical analysis for the worst-case conditional coverage based on the accuracy of the density estimation. While the assumptions are complex, the authors provide an interpretation for each assumption.
3.	The PFCP and GLCP methods have a valid marginal coverage guarantee, by building on conformal prediction.
4.	The paper is written clearly, presented well, and easy to follow.


## Weaknesses
1.	The proposed GLCP is very similar to the localized conformal prediction [1] and to distributional conformal prediction [2]. In contrast to the statement in line, 67, in my opinion, this extension is marginal and not novel.
2.	The experiments do not include the test-conditional miscoverage error of FedCP and FedCP-QQ.
3.	The experiments in [3] show that FedCP-QQ achieves a marginal coverage rate that is very close to $1-\alpha=90\%$ on the same real datasets. In contrast, the experiments in this paper show that the coverage rate attained by this method is far from the nominal level. This discrepancy between the two results is not addressed by the authors.
4.	The real-dataset experiments do not include a dataset that is suitable for federated learning setup, such as MedMNIST used in [4].



## References
[1] Leying Guan. Localized conformal prediction: A generalized inference framework for conformal prediction. Biometrika, 110(1):33–50, 2023

[2] Victor Chernozhukov, Kaspar Wüthrich, and Yinchu Zhu. Distributional conformal prediction. In Proceedings of the National Academy of Sciences, volume 118. National Acad Sciences, 2021.

[3] Pierre Humbert, Batiste Le Bars, Aurélien Bellet, and Sylvain Arlot. One-shot federated conformal prediction. In International Conference on Machine Learning, pages 14153–14177. PMLR, 2023

[4] Charles Lu and Jayasheree Kalpathy-Cramer. Distribution-Free Federated Learning with Conformal Predictions, 2022.

---

> ### Author Rebuttal · Authors · 2025-07-27
>
> We sincerely thank the reviewer for your thoughtful and encouraging assessment of our work. We are particularly grateful for your recognition of PFCP's theoretical and empirical advantages over existing methods, and the marginal guarantee of our methods under personalized federated setting, as well as your appreciation of our efforts to maintain clear presentation while tackling technically complex problems. Their insightful comments have helped us further improve the manuscript. Below we provide detailed responses to all points raised.
>
> **W1:**
> We appreciate the reviewer's insightful observation regarding the relationship between GLCP and existing localized conformal prediction methods.
> - We would like to clarify that GLCP was indeed designed to *generalize* and *unify* existing approaches.
> - When using kernel conditional CDF estimation (NW estimator), GLCP reduces to Localized Conformal Prediction (LCP). With identity score mapping, GLCP becomes equivalent to Distributional Conformal Prediction (DCP).
>
> We will revise and tone down our statement on GLCP accordingly in the revised version.
>
> **W2\&W4:**
> Thanks for the valuable feedback regarding reporting test-conditional miscoverage error of FedCP and FedCP-QQ, and including a dataset that is suitable for federated learning setup.
> - In response, we have supplemented table below with new real data results comparing test-conditional miscoverage rates, including evaluations on: (1) the DermaMNIST classification dataset from MedMNIST that is suitable for federated learning setup, and (2) two additional baselines (CPhet [1] and CPlab [2]).
> ||Dataset|GLCP|PFCP|FedCP|FedCP-QQ|CPhet|CPlab|
> |-|-|-|-|-|-|-|-|
> |Marginal|BIO|0.902|0.903|**0.993**$\times$|**0.970**$\times$|**0.981**$\times$|**0.978**$\times$|
> ||BIKE|0.900|0.900|**0.885**$\times$|**0.890**$\times$|**0.883**$\times$|**0.885**$\times$|
> ||CRIME|0.900|0.896|**0.866**$\times$|**0.852**$\times$|**0.864**$\times$|**0.863**$\times$|
> ||STAR|0.898|0.899|0.897|**0.910**$\times$|0.899|0.897|
> ||CONCRETE|0.903|0.905|**0.947**$\times$|**0.963**$\times$|**0.946**$\times$|**0.949**$\times$|
> ||DERMA|0.895|0.895|**0.830**$\times$|**0.819**$\times$|**0.849**$\times$|**0.867**$\times$|
> |Miscoverage|BIO|0.0315|**0.0199**(36.8%)|0.0941|0.0753|0.0844|0.0822|
> ||BIKE|0.0234|**0.0184**(21.4%)|0.0678|0.0647|0.0690|0.0681|
> ||CRIME|0.0387|**0.0268**(30.7%)|0.0426|0.0495|0.0429|0.0439|
> ||STAR|0.0392|**0.0243**(38.0%)|0.0500|0.0492|0.0496|0.0495|
> ||CONCRETE|0.0366|**0.0238**(35.0%)|0.0582|0.0675|0.0580|0.0600|
> ||DERMA|0.0307|**0.0264**(14.0%)|0.0939|0.1004|0.0802|0.0701|
> |Size|BIO|0.514|**0.503**(2.1%)|0.999|0.709|0.827|0.805|
> ||BIKE|4.097|**3.988**(2.7%)|4.004|4.091|3.967|3.996|
> ||CRIME|5.183|**4.496**(13.3%)|3.918|3.772|3.886|3.880|
> ||STAR|48.50|**43.02**(11.3%)|42.80|45.30|43.08|42.85|
> ||CONCRETE|34.49|**29.38**(17.7%)|34.71|38.97|34.69|35.15|
> ||DERMA|2.409|**2.327**(3.4%)|1.233|1.199|1.331|1.426|
>
> - Moreover, we would like to clarify that in our personalized federated learning setting with test data from the target agent distribution, ensuring valid marginal coverage is the fundamental requirement to be satisfied first, as it forms the basis for a reliable conformal prediction set. FedCP, FedCP-QQ, CPhet, and CPlab cannot guarantee this crucial marginal coverage property in our setting.
> - Although FedCP, FedCP-QQ, CPhet, and CPlab cannot guarantee marginal coverage, they may occasionally achieve the desired coverage level in certain scenarios (e.g., the STAR dataset). Even in these cases, GLCP and PFCP outperform the other methods in test-conditional miscoverage error.
>
> **W3:**
> We sincerely appreciate the detailed comment regarding this experimental detail.
> - We would like to clarify that FedCP-QQ requires all agents' data to be i.i.d. to achieve marginal coverage, which fundamentally contradicts our PFCP setting where the target agent and source agents follow different data distributions and the test point is from the target agent distribution.
> - In our experimental design, we artificially partition agents such that they follow distinct distributions to reflect the personalized federated learning scenario - while FedCP-QQ employs random sampling from the entire dataset for each agent in their experiments to reflect the i.i.d. case.
> - Thus, under our experiment setup, the distributional heterogeneity between target and source agents inherently violates FedCP-QQ's i.i.d. assumption and consequently undermines its marginal coverage guarantees.
>
> **Q1:**
> - The sentence in line 200 was originally intended to describe the existence of two different approaches to estimating $F(s\mid x)$. However, the use of the word "valid" may have been somewhat misleading. We meant they are both consistent estimators of $F(s\mid x)$ under certain conditions theoretically.
> - In the revised version, to avoid confusion, we will rephrase it as:
>
> "both $\mathbb{E}\_{S\sim\widehat{F}}\mathbb{1}(S\leq s)$ and $\mathbb{E}\_{S\sim\widehat{F}\_{\rm mix}(s\mid x)}\\{\mathbb{1}(S\leq s)\widehat{r}(S\mid x)\\}$ can serve as estimators of $F(s\mid x)$".
>
>
> **Q2:**
> We appreciate this insightful question on our theoretical results.
> - In Lemma 2.6, the first term in the right-hand side of the inequality represents the global calibration error, which is typically small and scales with the target agent's sample size $n$.
> When $\widehat{F}$ is the empirical distribution estimator, this term approximately equals $\epsilon + \delta + 1/n$.
> - The second term quantifies the estimation accuracy of the conditional distribution estimator, indicating that pointwise test-conditional coverage is governed by the local precision of the conditional distribution estimation.
> - Theorem 2.8 further establishes that the aggregated conditional distribution estimation is controlled by both the individual conditional distribution estimation accuracies and the precision of the density ratio estimation. In practice, source data is often abundant, enabling accurate estimation of its conditional distribution, in which case the overall precision becomes primarily determined by the density ratio estimation error. When sufficient relevance exists between source and target domains, this density ratio estimation can be performed with higher precision.
> - Even though it is unclear whether the bounds in Lemma 2.6 and Theorem 2.8 are sharp,  they are sufficient to provide theoretical justification for PFCP's improved performance compared to GLCP, which is the main focus of our theoretical study.
>
>
> **Q3:**
> Thank you for pointing out this typo. The integral should indeed be taken with respect to the score variable $s$ rather than the input variable $x$.
>
> **Q4:**
> - For synthetic data, the randomness is introduced through different random seeds when sampling from the given distributions.
> - For real-world datasets, the stochasticity arises from different random seeds used to partition the data into training, calibration, and test sets.
>
> **Q5:**
> - For synthetic data, we generate 100 prediction sets $C_1(x_i),...,C_{100}(x_i)$ for each test point $x_i$, compute the corresponding coverage using its true distribution we get $p_1(x_i),...,p_{100}(x_i)$, and estimate the conditional coverage at $x_i$ by averaging these values. The test-conditional miscoverage is then obtained by averaging across all test points.
> - For real-world data, we first partition the feature space into $k$ clusters using $k$-means. During each experimental repeat, we compute the empirical coverage within each cluster, then average these values across $100$ repeats to obtain the test-conditional coverage estimates. The final test-conditional miscoverage metric is calculated as a weighted average across all clusters, with weights proportional to the number of test samples in each cluster.
>
> **Minor1:**
> We sincerely thank the reviewer for identifying this omission. Lemma 2.1 follows directly from the basic validity guarantees of conformal prediction theory. In the revised version, we will add this citation.
>
> **Minor2:**
> Thank you for the suggestion to improve the clarity of our notation. In the revised manuscript, we will move the definition of the weight $\pi$ closer to its first usage in line 147.
>
> We are more than happy to answer any further questions during the Discussion.
>
> **Reference**
>
> [1] Vincent Plassier, Nikita Kotelevskii, Aleksandr Rubashevskii, Fedor Noskov, Maksim Velikanov, Alexander Fishkov, Samuel Horvath, Martin Takac, Eric Moulines, and Maxim Panov. Efficient conformal prediction under data heterogeneity. In International Conference on Artificial Intelligence and Statistics, pages 4879–4887. PMLR, 2024.
>
> [2] Vincent Plassier, Mehdi Makni, Aleksandr Rubashevskii, Eric Moulines, and Maxim Panov. Conformal prediction for federated uncertainty quantification under label shift. In International Conference on Machine Learning, pages 27907–27947. PMLR, 2023.

---

> > ### Comment · Reviewer_2MYP · 2025-08-02
> >
> > I thank the authors for their response and the additional experiments, which highlight the strengths of the proposed approach. I will maintain my score.

---

> > > ### Author Response · Authors · 2025-08-04
> > >
> > > We sincerely appreciate your time and valuable feedback, which helped us improve the paper. Thank you for maintaining your positive assessment of our work!

---

### Official Review · Reviewer_P7Ti · 2025-06-29

**Clarity:** 3
**Significance:** 2
**Originality:** 1
**Rating:** 4
**Confidence:** 4

**Summary:**

This paper introduces a conformal prediction method to quantify the uncertainty of algorithms in a personalized Federated Learning (FL) setting. An important distinction from prior works is that each agent has its own prediction function, a characteristic of the personalized FL setting. The authors claim to propose a method that 1) constructs conformal prediction sets for a target agent using data from all agents in a privacy-preserving manner 2) ensures marginal coverage and asymptotic conditional coverage. Finally, they show the efficiency of their method through various experiments.

**Questions:**

1\ As already said, Theorem 2.4 only depends on $n$ and not on all the data points. Hence, how do you theoretically know that your method uses all the points for the coverage?

2\ Since the article's contribution is mainly a new score function, what are the particularities of the techniques used for the proofs compared to other CP articles that don't deal with federated learning?

3\ It is say that the method is private but the privacy is never quantify. How do you know that your method is private? And are you able to give theoretical results about that?

4\ In line 186, it is said that « we obtain superior estimates that simultaneously improve conditional coverage accuracy and yield tighter prediction sets. » How do you know that since no theoretical results are given about the size of the sets ?

5\ In experiments, which score function is used for FedCP and FedCPQQ ?

**Ethical Concerns:**

["NO or VERY MINOR ethics concerns only"]

**Final Justification:**

The authors made a great effort in their responses to my questions.
After the rebuttal and based on the other reviews, I raised my score to 4.

**Limitations:**

yes

**Paper Formatting Concerns:**

Everything seems fine.

**Quality:**

2

**Strengths And Weaknesses:**

------- Strengths -------

1. The paper is well-written, and the subject is of significant interest to the conformal prediction community.

2. The method is clearly explained.

3. The related works section is thoroughly detailed.


------- Weaknesses -------

1. The article is really just an implementation of a new score function adapted to the federated learning framework.

2. The bound in Theorem 2.4 only depends on $n$, the number of points for the $K+1$-th agent. In contrast, in Theorem 2.2 of [1] for instance, the bound depends on $N$ the total number of points (i.e. the summation of the number of points of each agent).

3. Overall, it seems to me that the main contribution of this article is more about how to build CDF estimators in FL than how to perform conformal prediction in a personalized federated learning context (in a sens that the conformal part used only standard results).

4. (minor) The size of returned sets must always be indicated. For instance, this is not the case in Figure 1.

[1] Conformal Prediction for Federated Uncertainty Quantification Under Label Shift, Plassier et. al

---

> ### Author Rebuttal · Authors · 2025-07-27
>
> We thank the reviewer for the positive feedback regarding the paper's clarity, methodological presentation, and comprehensive literature review, as well as thoughtful and valuable suggestions. We address the key concerns below:
>
> **W1\&Q2\&W3:**
> We respectfully clarify that our contribution is *not* limited to introducing a new score function and disagree with the claim that our contribution is mainly about constructing CDF estimators.
> - First, we consider conformal prediction under the PFCP setting, where test data comes from the target agent, not the mixture of all source agents. Our framework not only guarantees marginal coverage on the target agent regardless of the source data distribution, but also improves test-conditional coverage for the target agent, which is an objective that existing federated conformal prediction methods have not addressed. This novel formulation, as noted by Reviewer kS3y, distinguishes our work from other FCP approaches.
> - Second, the fundamental motivation behind PFCP is to simultaneously guarantee marginal coverage while enhancing test-conditional coverage on the target agent. The key innovation resides in effectively transferring information from source agents to improve the conditional coverage performance on the target agent. This strategic information transfer, achieved through our carefully designed framework, is novel as noted by Reviewer kS3y.
> - Third, while the core technical contribution of our work lies in the construction of novel conformal scores, these scores are not simple function mappings. They are carefully designed fusion estimators that incorporate information from other source agents while strictly preserving the marginal coverage guarantee.
> - Our theoretical results go beyond standard CP analyses. In particular, Lemma 2.6 establishes a crucial theoretical connection between the CDF estimator and the test-conditional coverage of conformal prediction, which serves as the fundamental basis for analyzing the improvement. Further, Theorem 2.8 provides a non-asymptotic bound demonstrating that PFCP improves conditional coverage over using only local (target-agent) data (GLCP), thus justifying the benefit of leveraging source information through our reweighting scheme. This theoretical analysis proves PFCP attains superiority over GLCP and prior works, as noted by Reviewer 2MYP.
>
> **W2\&Q1:**
> Thanks for the question regarding the difference in sample size dependence between our work and [2]. The distinction stems from fundamentally different problem settings and objectives. Our PFCP framework has several key differences compared with [2]:
> - In PFCP setting, the test sample comes from the target agent, with distribution different from source agents, and we make minimal distributional assumptions: we only require the target agent's n samples to be i.i.d., without imposing any restrictive requirements between source and target agents, nor assuming known density ratios of any form. This is because we operate in a more challenging environment where source and target agents have different distributions (not limited to only label shift or covariate shift);
> - [2] focuses specifically on label shift scenarios where $X\mid Y$ distributions are identical across domains. Their Theorems 2.1 and 2.2 show that with *known* density ratios of $Y$, weighted conformal prediction (WCP) can leverage all $N$ samples. In practice (as noted in their Theorem 2.4), when density ratios must be estimated, marginal coverage error includes an additional term $R$ from estimation error, and this breaks the exact marginal coverage guarantee in our PFCP setting.
> - Our method provides strict finite-sample marginal coverage guarantees regardless of source agent data quality because we only rely on the target agent's $n$ samples for coverage calibration and source data only assists in improving conditional coverage. This is formalized in Theorem 2.8, which provides a non-asymptotic bound showing that our approach (PFCP) can improve conditional coverage compared to methods using only local data. The estimation of the density ratio $\hat{r}$, which is crucial for this improvement, depends on data from all source agents.
>
> **W4:**
> Thank you for pointing this out. We have now included the size of the prediction sets in the Table below. Since the other baseline methods fail to consistently achieve the marginal coverage requirement, we omit detailed size comparisons for these approaches. Our results demonstrate that the density ratio estimation using calibrated neural networks (PFCP2) yields significantly more efficient prediction sets compared to the logistic regression approach (PFCP1) and GLCP.
>
> |d|10|15|20|25|30|10|15|20|25|30|
> |-|-|-|-|-|-|-|-|-|-|-|
> ||||S1.1|||||S1.2|||
> |GLCP|**3.60**|**4.42**|**5.08**|**5.91**|**6.84**|**3.60**|**4.42**|**5.08**|**5.91**|**6.84**|
> |PFCP1|**3.76**|**3.97**|**4.28**|**4.88**|**5.90**|**4.23**|**4.12**|**4.32**|**5.02**|**5.76**|
> ||-4.34%|10.01%|15.73%|17.49%|13.79%|-17.52%|6.74%|14.95%|15.08%|15.75%|
> |PFCP2|**2.97**|**3.20**|**3.69**|**4.59**|**5.57**|**2.97**|**3.21**|**3.62**|**4.46**|**5.39**|
> ||17.38%|27.49%|27.28%|22.36%|18.57%|17.62%|27.22%|28.75%|24.54%|21.27%|
> ||||S1.1|||||S1.1||
> |GLCP|**4.28**|**4.88**|**5.29**|**6.16**|**7.15**|**4.28**|**4.88**|**5.29**|**6.16**|**7.15**|
> |PFCP1|**4.12**|**4.44**|**4.61**|**5.20**|**6.29**|**4.25**|**4.48**|**4.65**|**5.23**|**6.24**|
> ||3.64%|8.86%|12.87%|15.60%|11.94%|0.69%|8.13%|12.06%|15.14%|12.76%|
> |PFCP2|**3.61**|**3.90**|**4.17**|**5.05**|**6.08**|**3.57**|**3.87**|**4.07**|**4.84**|**5.90**|
> ||15.54%|20.03%|21.15%|18.02%|14.95%|16.45%|20.67%|23.08%|21.56%|17.41%|
>
> **Q3:**
> Thanks for raising this important point.
> - In our framework, privacy is maintained in the following ways: (i) no agent accesses other agents’ raw local data; (ii) information exchange only occurs during the estimation of $\widehat{F}_{\text{mix}}$ and the density ratio, both estimation optimized via the FedAvg algorithm; and (iii) the estimated CDF involves only conditional distribution of conformity scores which contains no invertible mapping to raw features or labels.
> - We acknowledge that although FedAvg is widely used, it may still carry risks of privacy leakage. However, we can directly replace FedAvg with existing privacy-preserving federated optimization techniques, such as differentially private mechanisms [3] or secure aggregation protocols [4], to provide formal privacy guarantees.
>
> We will add more discussion in the revised version and consider this a direction for future work.
>
> **Q4:**
> We appreciate the reviewer’s insightful question. It touches upon an important aspect of our method’s performance.
> - As stated in Theorem 2.8, our work provides a theoretical guarantee on the improvement of conditional coverage. However, we acknowledge that a formal theoretical result on the reduction of prediction set size is not currently provided.
> - Instead, we substantiate this claim through extensive empirical evidence from both simulation studies and real-world datasets, where PFCP consistently outperforms GLCP in producing smaller prediction sets. Please refer to Table 1 in the original manuscript, Figure 3 and Figures 16--19 in the appendix, as well as Table below and Table above provided in our response to W4, for detailed results supporting this observation.
>
> ||Dataset|GLCP|PFCP|FedCP|FedCP-QQ|CPhet|CPlab|
> |-|-|-|-|-|-|-|-|
> |Marginal|BIO|0.902|0.903|**0.993**$\times$|**0.970**$\times$|**0.981**$\times$|**0.978**$\times$|
> ||BIKE|0.900|0.900|**0.885**$\times$|**0.890**$\times$|**0.883**$\times$|**0.885**$\times$|
> ||CRIME|0.900|0.896|**0.866**$\times$|**0.852**$\times$|**0.864**$\times$|**0.863**$\times$|
> ||STAR|0.898|0.899|0.897|**0.910**$\times$|0.899|0.897|
> ||CONCRETE|0.903|0.905|**0.947**$\times$|**0.963**$\times$|**0.946**$\times$|**0.949**$\times$|
> ||DERMA|0.895|0.895|**0.830**$\times$|**0.819**$\times$|**0.849**$\times$|**0.867**$\times$|
> |Miscoverage|BIO|0.0315|**0.0199**(36.8%)|0.0941|0.0753|0.0844|0.0822|
> ||BIKE|0.0234|**0.0184**(21.4%)|0.0678|0.0647|0.0690|0.0681|
> ||CRIME|0.0387|**0.0268**(30.7%)|0.0426|0.0495|0.0429|0.0439|
> ||STAR|0.0392|**0.0243**(38.0%)|0.0500|0.0492|0.0496|0.0495|
> ||CONCRETE|0.0366|**0.0238**(35.0%)|0.0582|0.0675|0.0580|0.0600|
> ||DERMA|0.0307|**0.0264**(14.0%)|0.0939|0.1004|0.0802|0.0701|
> |Size|BIO|0.514|**0.503**(2.1%)|0.999|0.709|0.827|0.805|
> ||BIKE|4.097|**3.988**(2.7%)|4.004|4.091|3.967|3.996|
> ||CRIME|5.183|**4.496**(13.3%)|3.918|3.772|3.886|3.880|
> ||STAR|48.50|**43.02**(11.3%)|42.80|45.30|43.08|42.85|
> ||CONCRETE|34.49|**29.38**(17.7%)|34.71|38.97|34.69|35.15|
> ||DERMA|2.409|**2.327**(3.4%)|1.233|1.199|1.331|1.426|
>
> **Q5:**
> - Since all the datasets used in our main experiments are regression tasks, all methods (including FedCP and FedCPQQ) adopt the standard residual score function, defined as $s(x,y) = |y - f(x)|$, where $f$ is the pre-trained regression model.
> - For the classification task (DermaMNIST dataset) mentioned by Reviewer kS3y, we use the common classification score function $s(x,y) = 1 - [f(x)]_y$, where $f(x)$ returns the predicted class probabilities and $[f(x)]_y$ denotes the predicted probability of the true label $y$.
>
> We are more than happy to answer any further questions during the Discussion.
>
> **Reference**
>
> [1] Charles Lu, et al. (2023) Federated conformal predictors for distributed uncertainty quantification.
>
> [2] Vincent Plassier, et al. (2023) Conformal prediction for federated uncertainty quantification under label shift.
>
> [3] McMahan, H. B., et al. (2017). Learning differentially private recurrent language models.
>
> [4] Bonawitz, Keith, et al. (2017). Practical secure aggregation for privacy-preserving machine learning.

---

> > ### Comment · Reviewer_P7Ti · 2025-08-05
> >
> > Thank you for the detailed response.
> >
> > I remain skeptical about the answer to Q1. To achieve the same coverage bound, we can apply the CP locally, which seems to indicate that the calibration part does not use all available data and is therefore not “federated.”
> >
> > Furthermore, claiming that the algorithm is private without providing formal proof seems premature and potentially misleading. It is important to back up such claims with rigorous evidence.

---

> > > ### Author Response · Authors · 2025-08-05
> > >
> > > We appreciate the reviewer's engagement, but we respectfully disagree with the claim that our method is "not federated" simply because calibration uses only target-agent data.
> > > - Our method is designed for *personalized* federated learning, where the primary goal is to ensure strict marginal coverage guarantees on the target agent's test data (Theorem 2.4), while simultaneously improving test-conditional coverage through federated collaboration. This is fundamentally different from conventional federated conformal prediction that may sacrifice local guarantees for global consensus.
> > > - Our method does use all source agents' data ​for conditional distribution estimation, which improves test-conditional coverage (Theorem 2.8). This is the key federated learning component, as it leverages cross-agent information to enhance performance beyond what local data alone could achieve. Your suggestion to "apply CP locally" would fail to achieve this improvement, as local data is scarce and the conditional distribution is poorly estimated.
> > > - Moreover, marginal coverage guarantees require stronger distributional assumptions on calibration data, either exchangeable or known density ratios between source and target distributions. We would appreciate your perspective on why federated learning should prioritize calibration data scope over maintaining these foundational assumptions, while severe source-target distribution shifts could lead to catastrophic gaps in marginal coverage. Could you help us understand why​ 'federated' should be exclusively defined by calibration data usage, when our method achieves federated improvement in conditional coverage?
> > >
> > > We acknowledge the reviewer's concern about privacy but emphasize that our privacy claims are not  novel:
> > > - Our method inherits FedAvg's privacy properties, which are extensively studied in FL literature. As stated in our rebuttal, FedAvg can be directly replaced with differentially private FL or secure aggregation if formal guarantees are required. This is standard in federated learning, and we acknowledge that such mechanisms could be readily incorporated if needed for specific privacy requirements.
> > > - Moreover, we would appreciate your specific guidance on what type of privacy proof you expect.

---

> > > > ### Comment · Reviewer_P7Ti · 2025-08-05
> > > >
> > > > Thank you for your answer.
> > > >
> > > > Regarding privacy, one possibility would be to prove some results as in [1] and highlight how the privacy affects coverage, for example.
> > > >
> > > > [1] Private Prediction Sets, Private Prediction Sets, Anastasios N. Angelopoulos, Stephen Bates, Tijana Zrnic, Michael I. Jordan

---

> > > > > ### Author Response · Authors · 2025-08-05
> > > > >
> > > > > We appreciate your reference to [1], which introduces a privacy and accuracy trade-off where stronger protection (higher $\\epsilon$ in [1]) degrades marginal coverage guarantees. Therefore, [1] highlight how the privacy affects coverage. In contrast, our method avoids this limitation.
> > > > > - In our method, calibration uses only target-agent local data, which requires no privacy protection and preserves exchangeability by construction.
> > > > > - The federated steps (density ratio and conditional distribution estimation) operate under standard FL privacy paradigms. Whether using FedAvg (sharing model updates) or advanced methods like DP-FedAvg, the exchangeability of calibration data remains intact. This architectural separation ensures marginal guarantees theoretically uncompromised.
> > > > >
> > > > > We hope this clarification addresses the core distinction between our framework and [1]'s approach.
> > > > >
> > > > > [1] Angelopoulos, Anastasios Nikolas, et al. "Private prediction sets." Harvard Data Science Review 4.2 (2022).

---

> > > > > ### Author Response · Authors · 2025-08-06
> > > > >
> > > > > We appreciate the opportunity to clarify our theoretical guarantees and privacy property. Would you have any further questions on these points? We are happy to provide additional clarification if needed. Your feedback has been valuable in refining this work.

---

> > > > > > ### Comment · Reviewer_P7Ti · 2025-08-06
> > > > > >
> > > > > > Yes, thank you very much for your responses.

---

> > > > > > > ### Author Response · Authors · 2025-08-06
> > > > > > >
> > > > > > > Thank you for your prompt response. To ensure we have fully addressed your concerns, may we kindly ask:
> > > > > > > - Regarding the federated aspect, does our clarification about target-only calibration (for marginal guarantees) and federated density ratio estimation (for conditional improvement) resolve your original skepticism?
> > > > > > > - On privacy guarantees, would you like us to provide additional formal analysis or explore alternative privacy proofs?
> > > > > > >
> > > > > > > Your further guidance would be invaluable in helping us make the most appropriate revisions.

---

> > > > > > > > ### Comment · Reviewer_P7Ti · 2025-08-09
> > > > > > > >
> > > > > > > > This is ok for me. Thank you for your answers.

---

> > > > > > > > > ### Author Response · Authors · 2025-08-09
> > > > > > > > >
> > > > > > > > > Thank you for your positive feedback. We truly appreciate your time and constructive engagement throughout the review process.

---

### Official Review · Reviewer_Tp6A · 2025-07-03

**Clarity:** 3
**Significance:** 3
**Originality:** 3
**Rating:** 4
**Confidence:** 3

**Summary:**

The authors propose PFCP (Personalized Federated Conformal Prediction), which aims to combine personalized federated learning with conformal prediction. This can provide uncertainty sets that are both agent-specific and instance-specific. The authors also introduce GLCP (Generalized Localized Conformal Prediction) an extension of personalized federated learning that can accommodate any conditional distribution estimator. The proposed method PFCP shows how to leverage data from source agents to improve prediction sets --compared to GLCP which can be excessively large-- for a target agent while maintaining privacy and marginal coverage guarantees.

**Questions:**

Please address Weaknesses W1, W2, and W3.

**Ethical Concerns:**

["NO or VERY MINOR ethics concerns only"]

**Final Justification:**

The authors addressed my concerns about the lacking 2 baselines, I appreciate their experimental setup which seems modular enough to run things easily during the rebuttal phase. I think this is technically strong and I don't have other remaining pressing concerns.
That said, I don't think this paper would have a high impact on the field of conformal prediction nor a moderate impact on multiple areas of AI, hence I didn't recommend 5.

**Limitations:**

yes

**Quality:**

3

**Strengths And Weaknesses:**

Strengths:
1.  Combining both personalized federated learning and conformal prediction addresses an important field.
2.  The paper provides formal marginal coverage guarantees for their method.
3. The paper is well written and easy to follow.

Weaknesses:
1. The authors mention that the works of Plassier et al. [27, 28] are the most closely related works. However, they do not compare against these important baslines. Could the authors highlight the key differences of why they are not in the comparison table? Otherwise, could they add the results of these methods in Table 1?
2. Limited discussion of when the method might fail. Could the authors expand on this?
3. I am not sure if "personalized" federated learning is the proper term to be used in this context. This method only improves the prediction sets for the target agent satisfying Assumption 2. Could the authors reflect on this?

[27] Vincent Plassier, Alexander Fishkov, Maxim Panov, and Eric Moulines. Conditionally valid probabilistic conformal prediction. stat, 1050:1, 2024.
[28] Vincent Plassier, Nikita Kotelevskii, Aleksandr Rubashevskii, Fedor Noskov, Maksim Velikanov, Alexander Fishkov, Samuel Horvath, Martin Takac, Eric Moulines, and Maxim Panov. Efficient conformal prediction under data heterogeneity. In International Conference on Artificial Intelligence and Statistics, pages 4879–4887. PMLR, 2024.

---

> ### Author Rebuttal · Authors · 2025-07-27
>
> We are grateful for your positive perspective recognizing the importance of combining personalized federated learning with conformal prediction, as well as our method's formal marginal coverage guarantees and clear presentation. We are grateful to the reviewer for the insightful and constructive feedback. These comments have significantly contributed to improving the quality and presentation of our work. Below we provide detailed responses and clarifications to address the raised concerns.
>
> **W1:**
> We appreciate this important suggestion.
> - First, we would like to clarify a typo in our citation in line 96. We mistakenly cited [1,2], but it should have been [1] and [3], which correspond to their approaches under label shift and covariate shift settings, respectively. We denote these two methods as CPhet [3] and CPlab [1]. They share similarities with our approach but are designed for very different settings.
> - They are not compared because they cannot guarantee marginal coverage under our setup when the test data is from the target agent distribution, which is different from the source agents'.
> - We have now included these baselines in our experiments. Real data experiment results are shown in the table below:
>
> ||Dataset|GLCP|PFCP|FedCP|FedCP-QQ|CPhet|CPlab|
> |-|-|-|-|-|-|-|-|
> |Marginal|BIO|0.902|0.903|**0.993**$\times$|**0.970**$\times$|**0.981**$\times$|**0.978**$\times$|
> ||BIKE|0.900|0.900|**0.885**$\times$|**0.890**$\times$|**0.883**$\times$|**0.885**$\times$|
> ||CRIME|0.900|0.896|**0.866**$\times$|**0.852**$\times$|**0.864**$\times$|**0.863**$\times$|
> ||STAR|0.898|0.899|0.897|**0.910**$\times$|0.899|0.897|
> ||CONCRETE|0.903|0.905|**0.947**$\times$|**0.963**$\times$|**0.946**$\times$|**0.949**$\times$|
> ||DERMA|0.895|0.895|**0.830**$\times$|**0.819**$\times$|**0.849**$\times$|**0.867**$\times$|
> |Miscoverage|BIO|0.0315|**0.0199**(36.8%)|0.0941|0.0753|0.0844|0.0822|
> ||BIKE|0.0234|**0.0184**(21.4%)|0.0678|0.0647|0.0690|0.0681|
> ||CRIME|0.0387|**0.0268**(30.7%)|0.0426|0.0495|0.0429|0.0439|
> ||STAR|0.0392|**0.0243**(38.0%)|0.0500|0.0492|0.0496|0.0495|
> ||CONCRETE|0.0366|**0.0238**(35.0%)|0.0582|0.0675|0.0580|0.0600|
> ||DERMA|0.0307|**0.0264**(14.0%)|0.0939|0.1004|0.0802|0.0701|
> |Size|BIO|0.514|**0.503**(2.1%)|0.999|0.709|0.827|0.805|
> ||BIKE|4.097|**3.988**(2.7%)|4.004|4.091|3.967|3.996|
> ||CRIME|5.183|**4.496**(13.3%)|3.918|3.772|3.886|3.880|
> ||STAR|48.50|**43.02**(11.3%)|42.80|45.30|43.08|42.85|
> ||CONCRETE|34.49|**29.38**(17.7%)|34.71|38.97|34.69|35.15|
> ||DERMA|2.409|**2.327**(3.4%)|1.233|1.199|1.331|1.426|
>
>
>
> **W2:**
> Thank you for highlighting this.
> - Under our setup with test data from the target agent distribution, both GLCP and PFCP consistently guarantee the desired marginal coverage regardless of the distribution of target and source agents.
> - In contrast, FedCP, FedCP-QQ, CPhet and CPlab do not maintain such guarantees when the target distribution differs from the source agents’.
> - For the performance of test-conditional coverage, we outline the limitations of PFCP as follow:
>   - As shown in Theorem 2.8, achieving better conditional coverage than GLCP requires a moderately well-trained federated conditional density estimator, such that $\delta_1(x,\hat{F}_{\rm agg})<\delta_1(x,\hat{F})$, thereby leading to improved conditional coverage. However, if the density ratio is poorly estimated, the aggregation may not result in any improvement.
>   - If the source distribution significantly differs from the target distribution, for instance the support of $P$ and any $P_k$ do not intersect, the information from source agents cannot help improve the conformal set on the target agent.
>   - When the data on the target agent is extremely limited, insufficient to yield a reliable estimate of the conditional distribution $F(s\mid x)$ — both GLCP and PFCP may not deliver conformal sets with satisfying test-conditional coverage, though PFCP may still outperform GLCP in the sense of test-conditional coverage. Nonetheless, the marginal coverage remains valid in both cases.
>
> We'll add these discussions in the revised version.
>
>
> **W3:**
> We thank the reviewer for raising this important point.
> - Personalized setting refers to scenarios where agent-specific feature-label relationships differ substantially. The key characteristic of our problem is that test samples originate from a single designated target agent, and our primary objective is to guarantee the validity of conformal prediction specifically for this target agent. Our coverage guarantees are explicitly personalized to the target agent's distribution and simultaneously improving local (instance-specific) predictive performance.
> - Indeed, Assumption 2 serves as a condition to ensure the transferability of information from source to target domains. The fundamental requirement is that the conditional distribution $f(s\mid x)$ in the target domain can be effectively reconstructed through weighted combinations of source conditional distributions $\sum_k\pi_kf_k(s\mid x)$.
> - While Assumption 2 assumes a uniform lower bound on the density ratio $\frac{f(s\mid x)}{f_k(s\mid x)}$ for all $(s,x)$ pairs and some source distribution $P_k$, the actual necessary condition is substantially weaker. What we truly require is that the mixture of source distributions contains sufficient information about the target, meaning the corresponding density ratio $\sum_k\pi_kf_k(s\mid x)/f(s\mid x)$ has lower bounded (equation between line 560-561). This much milder condition would still guarantee that the target conditional distribution can be adequately approximated through our federated estimation framework, while being more realistic in practical applications. We will add more discussion on Assumption 2 in the revised version.
>
> We are more than happy to answer any further questions during the Discussion.
>
>  **Reference**
>
> [1] Vincent Plassier, Mehdi Makni, Aleksandr Rubashevskii, Eric Moulines, and Maxim Panov. Conformal prediction for federated uncertainty quantification under label shift. In International Conference on Machine Learning, pages 27907–27947. PMLR, 2023.
>
> [2] Vincent Plassier, Alexander Fishkov, Maxim Panov, and Eric Moulines. Conditionally valid probabilistic conformal prediction. stat, 1050:1, 2024.
>
> [3] Vincent Plassier, Nikita Kotelevskii, Aleksandr Rubashevskii, Fedor Noskov, Maksim Velikanov, Alexander Fishkov, Samuel Horvath, Martin Takac, Eric Moulines, and Maxim Panov. Efficient conformal prediction under data heterogeneity. In International Conference on Artificial Intelligence and Statistics, pages 4879–4887. PMLR, 2024.

---

> > ### Comment · Reviewer_Tp6A · 2025-08-04
> >
> > I appreciate the response of the authors, it has addressed most of my concerns.
> > I have raised my score to 4.

---

> > > ### Author Response · Authors · 2025-08-04
> > >
> > > Thank you for your constructive feedback and for raising your score​. We sincerely appreciate the time you took to review our work and are glad that our revisions addressed your concerns!

---

### Official Review · Reviewer_kS3y · 2025-07-03

**Clarity:** 3
**Significance:** 4
**Originality:** 3
**Rating:** 4
**Confidence:** 4

**Summary:**

Authors propose a personalization framework for  (conditional) conformal prediction in federated settings using engression and federation of a learnable conditional density estimator.

**Questions:**

* Why the proposed algorithm do not operate on classification datasets used in [1]. This limits the credibility of the algortihm as the FedCP is not tested on these datasets as well.
* Some very closest works mentioned [1,2] , but not included in the baselines. What is the exact reason? Some of the datasets in this study has been used
* What distribution shift do you consider in the proposed algorithm?


[1]  Charles Lu, Yaodong Yu, Sai Praneeth Karimireddy, Michael Jordan, and Ramesh Raskar. Federated conformal predictors for distributed uncertainty quantification. In International Conference on Machine Learning, pages 22942–22964. PMLR, 2023.

[2]Vincent Plassier, Alexander Fishkov, Maxim Panov, and Eric Moulines. Conditionally valid probabilistic conformal prediction. stat, 1050:1, 2024.

[3] Vincent Plassier, Nikita Kotelevskii, Aleksandr Rubashevskii, Fedor Noskov, Maksim Velikanov, Alexander Fishkov, Samuel Horvath, Martin Takac, Eric Moulines, and Maxim Panov. Efficient conformal prediction under data heterogeneity. In International Conference on Artificial Intelligence and Statistics, pages 4879–4887. PMLR, 2024

**Ethical Concerns:**

["NO or VERY MINOR ethics concerns only"]

**Final Justification:**

The authors provided reasonable arguments to my responses.

**Limitations:**

Yes.

**Paper Formatting Concerns:**

No concerns.

**Quality:**

3

**Strengths And Weaknesses:**

**Strengths**

* The personalization problem of federated conformal prediction is novel and interesting.
* Leveraging a federated conditional distribution estimator to leverage PFCP is novel.

**Weaknesses**
* Evaluation with unknown datasets compared to the baselines[1].
* Some missing baselines [2,3]
* GLCP is not clear. Is it also federated, or is there no communication between the clients in this case?
* As long as coverages are greater then 1 - $\alpha$, the coverage results are fine. However, the effect of personalization over the set size as opposed to [1] is not clear to me.
* The personalization part is not clear throughout the steps in lines 172-180. Is personalization done over the aggregation level , or not sharing some specific model?
* Notation is overall hard to follow as opposed to previous works in federated CP.

[1]  Charles Lu, Yaodong Yu, Sai Praneeth Karimireddy, Michael Jordan, and Ramesh Raskar. Federated conformal predictors for distributed uncertainty quantification. In International Conference on Machine Learning, pages 22942–22964. PMLR, 2023.

[2]Vincent Plassier, Alexander Fishkov, Maxim Panov, and Eric Moulines. Conditionally valid probabilistic conformal prediction. stat, 1050:1, 2024.

[3] Vincent Plassier, Nikita Kotelevskii, Aleksandr Rubashevskii, Fedor Noskov, Maksim Velikanov, Alexander Fishkov, Samuel Horvath, Martin Takac, Eric Moulines, and Maxim Panov. Efficient conformal prediction under data heterogeneity. In International Conference on Artificial Intelligence and Statistics, pages 4879–4887. PMLR, 2024

---

> ### Author Rebuttal · Authors · 2025-07-27
>
> Thank you for your positive assessment recognizing both the novelty of the personalization problem in federated conformal prediction and the methodological innovation of our federated conditional distribution estimator for PFCP. We sincerely thank the reviewer for the constructive and thoughtful feedback. Below, we address each concern in detail and provide clarifications and additional numerical results.
>
> **W1\&Q1\&W2\&Q2:**
> - First, we would like to clarify a typo in our citation in line 96. We mistakenly cited [3], and the correct references should be [2] (for label shift) and [4] (for covariate shift). We will correct this in the revised version.
> - For the classification experiments (W1\&Q1), we have now implemented the code for classification task using the score function $s(x,y)=1-[f(x)]_y$ as in [1]. We conduct additional experiments on DermaMNIST as a representative classification dataset (denoted as DERMA), and will include these new results in our revised manuscript.
> - For the additional baselines (W2\&Q2), we sincerely appreciate the reviewer's suggestion to include additional baselines. While [2] (CPlab) and [4] (CPhet) share conceptual similarities with our approach, we note they were originally designed for different problem settings. We have implemented both methods for fair comparison.
>  - The real data results are shown in Table below. The experimental results demonstrate that both GLCP and PFCP consistently achieve valid marginal coverage on the target agent, while the other baseline methods frequently fail to maintain this fundamental guarantee. Furthermore, PFCP exhibits significantly lower test-conditional miscoverage compared to GLCP, with both proposed methods substantially outperforming all remaining baselines.
>
> ||Dataset|GLCP|PFCP|FedCP|FedCP-QQ|CPhet|CPlab|
> |-|-|-|-|-|-|-|-|
> |Marginal|BIO|0.902|0.903|**0.993**$\times$|**0.970**$\times$|**0.981**$\times$|**0.978**$\times$|
> ||BIKE|0.900|0.900|**0.885**$\times$|**0.890**$\times$|**0.883**$\times$|**0.885**$\times$|
> ||CRIME|0.900|0.896|**0.866**$\times$|**0.852**$\times$|**0.864**$\times$|**0.863**$\times$|
> ||STAR|0.898|0.899|0.897|**0.910**$\times$|0.899|0.897|
> ||CONCRETE|0.903|0.905|**0.947**$\times$|**0.963**$\times$|**0.946**$\times$|**0.949**$\times$|
> ||DERMA|0.895|0.895|**0.830**$\times$|**0.819**$\times$|**0.849**$\times$|**0.867**$\times$|
> |Miscoverage|BIO|0.0315|**0.0199**(36.8%)|0.0941|0.0753|0.0844|0.0822|
> ||BIKE|0.0234|**0.0184**(21.4%)|0.0678|0.0647|0.0690|0.0681|
> ||CRIME|0.0387|**0.0268**(30.7%)|0.0426|0.0495|0.0429|0.0439|
> ||STAR|0.0392|**0.0243**(38.0%)|0.0500|0.0492|0.0496|0.0495|
> ||CONCRETE|0.0366|**0.0238**(35.0%)|0.0582|0.0675|0.0580|0.0600|
> ||DERMA|0.0307|**0.0264**(14.0%)|0.0939|0.1004|0.0802|0.0701|
> |Size|BIO|0.514|**0.503**(2.1%)|0.999|0.709|0.827|0.805|
> ||BIKE|4.097|**3.988**(2.7%)|4.004|4.091|3.967|3.996|
> ||CRIME|5.183|**4.496**(13.3%)|3.918|3.772|3.886|3.880|
> ||STAR|48.50|**43.02**(11.3%)|42.80|45.30|43.08|42.85|
> ||CONCRETE|34.49|**29.38**(17.7%)|34.71|38.97|34.69|35.15|
> ||DERMA|2.409|**2.327**(3.4%)|1.233|1.199|1.331|1.426|
>
> **W3:**
> As stated in line 116 of our paper, GLCP relies solely on the target agent's data, meaning that it does not involve any communication or data sharing with other agents. As we mentioned in line 139, this reliance on limited target data can lead to unreliable or excessively large prediction sets. This observation motivates our proposed PFCP method, which leverages data from other agents to improve efficiency.
>
> **W4:**
> We sincerely thank the reviewer for raising the important question regarding the coverage and prediction set size in the personalized setting.
> - [1] consider a setting where the test data is drawn from a mixture of source agents’ distributions—implicitly assuming a certain relationship between the source and target distributions. In contrast, we are considering the scenario when the test data is from a specific (target) agent. As a result, the method of [1] does not guarantee marginal coverage under our setting, which is also confirmed empirically in Figure 1 and Table 1.
> - The results in the above Table reveal an important trade-off: when baseline methods achieve marginal coverage exceeding the nominal level (0.9 in our experiments), their prediction sets become substantially larger than those produced by PFCP. Conversely, when these baselines fail to maintain the required coverage (falling below 0.9), their prediction sets shrink to unrealistically small sizes. However, such under-coverage scenarios render the size comparisons meaningless, as the resulting prediction sets lack the fundamental validity guarantee that forms the basis of conformal prediction.
> - In our proposed PFCP framework, personalization is achieved by leveraging information from source agents through density-ratio weighting, tailored for a specific target agent. This allows for a more accurate estimation of $F(s\mid x)$ compared to GLCP, which relies solely on target agent data. Consequently, PFCP leads to more efficient prediction intervals (i.e., smaller set sizes), as shown in Table 1.
>
> **W5:**
> We sincerely thank the reviewer for pointing out the lack of clarity in our explanation.
> - The personalization in our method is primarily achieved through the calibration step (Step 4) is performed on the target agent's data to guarantee marginal coverage on the target distribution.
> - The incorporation of source data is solely for improving the CDF estimation accuracy. The density ratio estimation emerges as a necessary component precisely because we introduce source data to enhance the conditional distribution estimation while maintaining the personalized calibration.
> - This design ensures that while we leverage additional data from source agents to improve the quality of our conformal prediction sets, the marginal coverage on the target agent is strictly guaranteed.
>
> **W6:**
> We sincerely appreciate this feedback and apologize for any confusion caused by our notation.
> - The complexity arises from the need to precisely represent multiple algorithmic components in our generalized framework. We apologize if the complexity of the notation caused any confusion.
> - To better address your concerns, could you kindly specify which particular notational aspects you found challenging to follow? This would enable us to provide more targeted explanations or improvements to the relevant notations.
>
> **Q3:**
> We sincerely thank the reviewer for highlighting this important aspect.
> - Our algorithm is designed for any type of heterogeneous local data distributions across agents.
> - In our synthetic data experiments, we primarily focus on concept shift, i.e., changes in $P_{Y\mid X}$ across agents, as described in Section 3.1.
> - For the real-world datasets, both covariate and concept heterogeneity exist among agents.
>
> We are more than happy to answer any further questions during the Discussion.
>
> **Reference**
>
> [1] Charles Lu, Yaodong Yu, Sai Praneeth Karimireddy, Michael Jordan, and Ramesh Raskar. Federated conformal predictors for distributed uncertainty quantification. In International Conference on Machine Learning, pages 22942–22964. PMLR, 2023.
>
> [2] Vincent Plassier, Mehdi Makni, Aleksandr Rubashevskii, Eric Moulines, and Maxim Panov. Conformal prediction for federated uncertainty quantification under label shift. In International Conference on Machine Learning, pages 27907–27947. PMLR, 2023.
>
> [3] Vincent Plassier, Alexander Fishkov, Maxim Panov, and Eric Moulines. Conditionally valid probabilistic conformal prediction. stat, 1050:1, 2024.
>
> [4] Vincent Plassier, Nikita Kotelevskii, Aleksandr Rubashevskii, Fedor Noskov, Maksim Velikanov, Alexander Fishkov, Samuel Horvath, Martin Takac, Eric Moulines, and Maxim Panov. Efficient conformal prediction under data heterogeneity. In International Conference on Artificial Intelligence and Statistics, pages 4879–4887. PMLR, 2024.

---

> > ### Comment · Reviewer_kS3y · 2025-08-05
> >
> > I have raised my score to 4. The authors' rebuttal helped to address most of my concerns.

---

> > > ### Author Response · Authors · 2025-08-05
> > >
> > > Thank you for raising your score​ and for acknowledging our efforts in addressing your concerns. We sincerely appreciate your constructive feedback, which has helped improve the paper!

---

### Decision · Program_Chairs · 2025-09-17

**Decision:**

Accept (poster)

**Comment:**

This paper presents a personalized federated conformal prediction method which can provide agent-specific and instance-localized uncertainty quantification. The work comes with finite-sample marginal coverage guarantees, as well as extensive experiments across synthetic and real-world datasets.

After initial reviews and discussions, the four reviewers unanimously praised the novelty and significance of this work in addressing an interesting problem of reliable uncertainty estimates in heterogeneous federated environments. Some minor concerns regarding presentation clarity, privacy claims, and experimental setups were effectively resolved in rebuttals and reviewer-author discussions. So the resulting sentiment is that this is a novel and technically sound FL paper worthy to be accepted to NeurIPS 2025. The authors are encouraged to address the issues raised in the reviews as much as possible in the revised paper.